# The Owls Are Not What They Seem: Health, Mood, and Sleep Problems Reported by Morning and Evening Types with Atypical Timing of Weekend Sleep

**DOI:** 10.3390/clockssleep7030035

**Published:** 2025-07-11

**Authors:** Arcady A. Putilov, Evgeniy G. Verevkin, Dmitry S. Sveshnikov, Zarina V. Bakaeva, Elena B. Yakunina, Olga V. Mankaeva, Vladimir I. Torshin, Elena A. Trutneva, Michael M. Lapkin, Zhanna N. Lopatskaya, Roman O. Budkevich, Elena V. Budkevich, Natalya V. Ligun, Alexandra N. Puchkova, Vladimir B. Dorokhov

**Affiliations:** 1Independent Research Group for Biomedical Systems Math-Modeling, 12489 Berlin, Germany; 2Department of Normal Physiology, Medical Institute of the Peoples’ Friendship University of Russia, 117198 Moscow, Russia; 3Department of Physiology, Ryazan State Medical University, 390026 Ryazan, Russia; 4Department of Physiology, Medical Institute of the Surgut State University, 628403 Surgut, Russia; 5Laboratory of Nanobiotechnology and Biophysics, North-Caucasus Federal University, 355029 Stavropol, Russia; 6Laboratory of Sleep/Wake Neurobiology, Institute of Higher Nervous Activity and Neurophysiology of the Russian Academy of Sciences, 117865 Moscow, Russia

**Keywords:** morningness-eveningness, trait- and state-like individual difference, self-rated health, depressed mood, sleep problems, simulation

## Abstract

Morningness-eveningness is usually assessed as either a trait or a state using either a morning–evening preference scale or sleep timing reported for free days, respectively. These assessments were implemented in numerous studies exploring the associations between morningness-eveningness and health, mood, and sleep problems. Evening types almost always had more problems than morning types. We examined these associations in university students with conflicting results of trait and state assessments of morningness-eveningness and tried to confirm their chronotype using a multidimensional chronotyping approach that recognizes four types other than morning and evening (lethargic, vigilant, napping, and afternoon). The conflicting trait and state assessments of morningness-eveningness were found in 141 of 1582 students. Multidimensional chronotyping supported morningness of morning types with late weekend sleep timing, and the associations with health, mood, and sleep problems resembled the associations of other morning types (i.e., these associations persisted despite late sleep timing). In contrast, evening types with early weekend sleep timing were more likely classified as lethargic or napping types rather than evening types. They did not resemble evening types in their associations with health, mood, and sleep problems (i.e., early sleep timing did not change these associations). Model-based simulations of the sleep–wake cycles of students with conflicting trait and state assessments suggested that their bedtimes cannot be solely determined by their biological clocks. On weekdays or weekends, mind-bedtime procrastination can lead to missing the bedtime signal from their biological clocks (i.e., self-deprivation of sleep or, in other words, voluntary prolongation of the wake phase of the sleep–wake cycle).

## 1. Introduction

The structure of individual variation in the subjectively assessed characteristics of daily rhythms appears to be multidimensional [1,2,3,4,5,6,7,8,9,10]. However, the variation in only one of several dimensions known under the term “morningness-eveningness” [11] is usually of interest for studies of individual differences in sleep–wake behavior and habits [12,13,14,15]. In the framework of such a unidimensional approach, an individual can be assigned to one of two poles of this morningness-eveningness dimension and named either a morning or evening (chrono)type [16] or nicknamed either a lark or owl, respectively. Usually, morningness-eveningness is assessed as a trait using a morning–evening preference scale (e.g., [17,18,19]). Consequently, individuals occupying the poles of this continuous preference scale can be named morning and evening types. Despite this most popular trait-like approach to morningness-eveningness assessments, it is also often unidimensionally assessed as a state rather than a trait using data on current sleep times on free days (e.g., [20,21,22,23]).

The correlations between trait and state assessments (i.e., between a preference score and a current sleep timing) are expected to be positive in sign and substantial in magnitude. Therefore, the cases of discrepancy between these assessments are relatively rare [24,25,26]. However, such exceptions cannot be fully excluded, and, therefore, it can be expected that there are individuals with discordant (conflicting) trait and state assessments of morningness-eveningness, i.e., who are classified as evening types based on their trait assessments and as morning types according to their state assessments or, conversely, who are classified as morning types according to their trait assessments and as evening types based on their state assessments. Moreover, neither trait nor state assessments can fully replace the objective measurements of the circadian phase of the rhythms marking the central circadian clock. It was shown that the trait and state assessments of chronotype correlate with the current circadian phase of the melatonin rhythm [27,28], but neither correlation was found to be strong [27,28], and chronotype categorizations based on either trait or state assessments demonstrated low concordance with that based on the circadian phase measurement [28].

The questionnaire tools for either trait or state assessments of morningness-eveningness were implemented in a large number of studies testing the links of chronotype with health (e.g., [29,30,31,32,33,34,35]), mood (e.g., [36,37,38,39,40,41,42,43,44,45,46]), and sleep problems (e.g., [47,48,49,50,51,52,53,54,55,56,57,58,59,60,61,62]). The results of the vast majority of studies suggested that such problems were more frequently associated with evening than morning types.

If morningness-eveningness is assessed as a trait-like characteristic of an individual, it is expected to be stable, long-lasting, and internally caused; conversely, if it is assessed as a state-like characteristic, it can be temporal, short-lasting, and caused by external circumstances. Given that current sleep timing belongs to the state-like characteristics of chronotypes, the questions arise: Can interventions leading to an advance of sleep timing improve health, mood, and sleep in evening types, and can interventions delaying sleep timing compromise health, mood, and sleep in morning types? Due to intercorrelation between trait and state assessments, it is difficult to determine whether the associations between chronotype and these problems are internal or external in nature. Therefore, it is difficult to evaluate the prospects of improving health, mood, and sleep by correcting sleep timing, a state-like characteristic of morningness-eveningness. Light can be shed on the issue of the internal vs. external nature of these associations in a study of individuals with discordant trait and state assessments of morningness-eveningness. If such associations are internally caused, the problems with health, mood, and sleep are expected to be more likely reported by evening types with early sleep timing rather than morning types with late sleep timing. Alternatively, these problems are expected to be more likely reported by morning types with late sleep timing rather than evening types with early sleep timing in the case of externally caused associations.

The conflicting trait and state assessments of morningness-eveningness can be, at least partly, explained by limitations of the unidimensional approach to categorization of chronotypes, i.e., when only two—either evening or morning—types can be distinguished at the poles of a preference scale. The application of a multidimensional approach to chronotyping can suggest that individuals with discordant trait and state assessments belong to other than just two types, morning and evening. For example, LIVEMAN chronotyping was introduced to distinguish between six types rather than two chronotypes (lethargic, vigilant, afternoon, and napping types in addition to morning and evening types) [6]. Consequently, these four other chronotypes can differ from morning and evening types in their pattern of associations with health, mood, and sleep problems. The difference in such associations can also be expected for those individuals with discrepant trait and state assessments of morningness-eveningness who were reclassified as lethargic, vigilant, afternoon, or napping types by means of this multidimensional chronotyping. Therefore, the associations with health, mood, and sleep problems of the individuals with discordant trait and state assessments of morningness-eveningness can resemble the associations of either morning or evening types or other types of the multidimensional classification of chronotypes.

Here, the associations between chronotype and health, mood, and sleep problems were explored in 1582 university students aged between 17 and 25 years. Like people of younger and older ages, they wake up relatively early on weekdays. On the other hand, they differ from younger and older people in reporting the latest weekend sleep timing (e.g., [63,64]). It can be speculated that some lark-students sleep late on weekends in order to remain in sync with their peers who practice late sleep timing. Conversely, some owl-students might sleep earlier on weekends to reduce the extent of displacement of their sleep relative to their advanced circadian sleep phase on weekdays. However, it is difficult to imagine university students who wake up early on weekends. Therefore, it can be expected that their trait and state assessments of morningness-eveningness disagree due to applying the unidimensional approach to chronotyping. These owl-students can belong to types other than evening and morning, and, consequently, they can differ from both morning and evening types in their pattern of associations with health, mood, and sleep problems.

Here, we asked the following questions about chronotypes with discordant trait and state characteristics of morningness-eveningness: 1. How many such chronotypes are there? 2. Can multidimensional chronotyping confirm unidimensional categorization of these chronotypes as either morning or evening types? 3. Can the patterns of associations of these chronotypes with health, mood, and sleep problems resemble the patterns established for chronotypes with concordant trait and state characteristics? 4. Can a model of sleep–wake regulation indicate an additional regulator of sleep times underlying the discrepancy between trait and state characteristics?

The following hypotheses were tested: 1. Chronotypes with discordant trait and state characteristics of morningness-eveningness are rare compared to chronotypes with concordant characteristics. 2. Morning or evening types with late or early sleep timing, respectively, are neither morning nor evening types of the multidimensional classification of chronotype. 3. When such multidimensional classification suggests that they are neither morning nor evening types, their associations with health, mood, and sleep concerns are different from the associations of chronotypes with concordant characteristics of morningness-eveningness. 4. Sleep times in chronotypes with discordant trait and state characteristics of morningness-eveningness can be explained by accounting for an additional parameter of underlying regulating processes in model-based simulations of their sleep–wake cycles.

## 2. Results

### 2.1. Division of Subsamples into Three Trait, Three State, and Five Trait+State Types

The mean ages ± standard deviations of 439 male and 1133 female university students were 19.7 ± 2.0 and 19.4 ± 1.5 years, respectively.

Table 1 includes the results of assigning students to chronotypes with concordant and discordant trait and state characteristics of morningness-eveningness. It contains the criteria applied for such an assignment to unidimensional trait types (3), unidimensional state types (3), and, finally, unidimensional trait+state types (5: Mm-, Me-, Neither-, Em-, and Ee-type).

The whole sample was subdivided into three trait subsamples using the sums of scores on scales designed to assess morning and evening/night components of morningness-eveningness (12-item M and E scales of the SWPAQ named “Morning Lateness” and “Evening Lateness”, respectively, and 10-item MS and NW scales of the SWAT named “Morning Sleepability” and “Nighttime Wakeability”, respectively). Each of these four scales consists of an equal number of positively and negatively worded statements (six pairs in the SWPAQ and five pairs in the SWAT). The responses to these statements were assigned as either +1 or −1 to refer to eveningness or morningness, respectively. Because each statement describes the habits or behaviors in a way that highlights eveningness or morningness, respectively, the sums of scores on two morningness-eveningness scales of each questionnaire (M+E and MS+NW) allowed the subdivision of the whole sample into subsamples of morning and evening types (M- and E-types, respectively) characterized by a positive or negative score, respectively. Further subdivisions using the scores from the scales of two questionnaires resulted in three trait subsamples (M-type with both M+E and MS+NW ≤ 0, other type, and E-type with both M+E and MS+NW > 0). Similarly, we subdivided the whole sample according to weekend bed- and risetimes, either not later or later than 12 a.m. and 9 a.m. (approximately one-third or two-thirds of the whole sample, respectively). Applying such criteria to bed- and risetimes provided the possibility of subdividing the whole sample into three state subsamples (m-type with bedtime not later than 12 a.m. and risetime not later than 9 a.m., other type, and e-type with bedtime later than 12 a.m. and risetime later than 9 a.m.). Finally, the combination of the three trait and three state subdivisions based on scores and sleep times, respectively, resulted in the subdivision of the whole sample into five subsamples: (1) Mm-type with M+E and MS+NW ≤ 0, bedtime not later than 12 a.m. and risetime not later than 9 a.m. (concordant morning-type assessments); (2) Me-type with M+E and MS+NW ≤ 0, but bedtime later than 12 a.m. and risetime later than 9 a.m. (discordant morning-type assessments); (3) Neither-type; (4) Em-type with M+E and MS+NW > 0, but with bedtime not later than 12 a.m. and risetime not later than 9 a.m. (discordant evening-type assessments); and (5) Ee-type with M+E and MS+NW > 0, bedtime later than 12 a.m. and risetime later than 9 a.m. (concordant evening-type assessments).

As expected, the number of evening-type students was greater than the number of morning-type students. Also, as expected, the number of concordant morning and evening types was greater than the number of discordant morning and evening types (Table 1). The discordant trait and state characteristics of morningness-eveningness were reported by only 8.9% of the total sample of 1582 students. Every third of morning trait types reported late weekend sleep times (Me-type), while only every tenth of evening trait types reported early weekend sleep times (Em-type), 5.4% and 3.5% of the whole sample, respectively (Table 1). While morning-type students frequently reported late weekend sleep timing, evening-type students rarely reported early weekend sleep timing. The difference between male and female students in the division of five subsamples of trait+state types was non-significant (χ^2^_4_ = 4.6, *p* = 0.334).

### 2.2. Scale Scores and Sleep Times in Five Trait+State Types

Figure 1 illustrates the differences between five trait+state types (Mm-, Me-, Neither-, Em-, and Ee-types) in scores on morningness-eveningness scales (Figure 1A) and sleep times (Figure 1B). As expected, the extent of morningness-eveningness gradually increased in the following direction: Mm-type (extremely negative scores) → Me-type (negative scores) → Neither-type (near-zero scores) → Em-type (positive scores) → Ee-types (extremely positive scores; Figure 1A). Early weekend sleep timing (bed- and risetime not later than 12 a.m. and 9 a.m., respectively) was reported by Mm- and Em-types, and late weekend sleep timing (bed- and risetime later than 12 a.m. and 9 a.m., respectively) was reported by Me- and Ee-types (Figure 1B).
Figure 1Scores on morningness-eveningness scales and sleep times in five trait+state types. Scores on morningness-eveningness scales and sleep times on weekends (**B**) were used for the subdivision of the whole sample into subsamples of unidimensional trait and state types, with the following subdivision into subsamples of trait+state types: Mm-, Me-, Neither-, Em-, and Ee-types. (**A**) Scores on scales designed to assess morning and evening/night components of morningness-eveningness (two 12-item scales of SWPAQ, “Morning Lateness” and “Evening Lateness”, and two 10-item scales of SWAT, “Morning Sleepability” and “Nighttime Wakeability”); M+E and MS+NW: Sums of scores on two scales of the SWPAQ and SWAT. (**B**) Rise- and bedtimes (scale on the left and right sides, respectively) on weekdays and weekends in five trait+state types. See the results of the statistical comparisons of five trait+state types in Table 2 and Table 3.
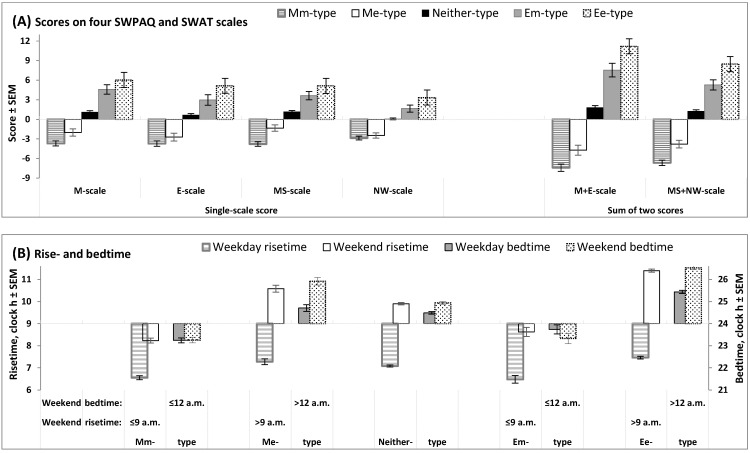


Similarity between the trait+state subdivisions was found for only one of six sleep times shown in Table 3, time in bed on weekends (the last column on the right). This similarity implies that the difference between the five trait+state types in sleep duration did not contribute to their difference in weekend sleep timing. Since weekday bedtimes in one of five subdivisions (Em-types) were, on average, somewhat earlier than weekend bedtimes in this subdivision (Figure 1B), it was necessary to include an additional parameter in the model simulating their sleep times (see Section 2.5).

### 2.3. Multidimensional Chronotyping of Trait, State, and Trait+State Types

The multidimensional approach to chronotyping provides the possibility to evaluate differences between individuals other than sleep phase, such as differences in afternoon and daily averaged levels of activity (afternoon vs. napping type and lethargic vs. vigilant type, respectively) [6]. Table 4 and Table 5 contain the results of the comparison of counts of chronotypes distinguished by applying single-dimensional and multidimensional approaches for chronotyping (columns and lines, respectively). The division of students into either three trait, three state, or five trait+state types is based on distinguishing between only two extreme chronotypes, morning or evening. In contrast, the multidimensional LIVEMAN chronotyping includes several other options, i.e., morning and evening types are only two of seven types (abbreviated as LIVEMAN types: Lethargic, Inconclusive, Vigilant, Evening, Morning, Afternoon, and Napping). The lethargic type has a low level of alertness in the morning, afternoon, and evening; the vigilant type has a high level in the morning, afternoon, and evening; the evening type has a low level in the morning, moderate in the afternoon, and high in the evening; the morning type has a high level in the morning, moderate in the afternoon, and low in the evening; the afternoon type has a low level in the morning, high in the afternoon, and moderate in the evening; the napping type has a high level in the morning, low in the afternoon, and moderate in the evening; and the inconclusive type is different from any of the six mentioned types (see also the notes for Table 4 and Appendix A). Therefore, a student assigned to the subsamples of morning types using single-dimensional chronotyping (i.e., morning trait type, morning state type, or morning trait+state type with either concordant or discordant trait and state assessments) can choose another of the morning-type options offered by the LIVEMAN classification of chronotypes. The results of LIVEMAN chronotyping of such morning types suggested that, nevertheless, morning-type students were less likely to choose other morning-type options provided by the LIVEMAN classification (Table 4 and Table 5). Therefore, none of the subsamples of morning types was an exception (Table 5).

Similarly, a student assigned to the subsamples of evening types using single-dimensional chronotyping (i.e., evening trait type, evening state type, or evening trait+state type with either concordant or discordant trait and state assessments) can choose another evening-type option offered by the LIVEMAN classification. The results of LIVEMAN chronotyping of such evening types suggested that they are more likely to choose other evening-type options from the LIVEMAN classification when they are an evening trait type, evening state type, or evening trait+state type with concordant trait and state assessments (Table 4 and Table 5). However, one of the subsamples of evening types with discordant trait+state assessments (Em-types) was an exception (Table 5).

Overall, these results suggested that a student assigned to a subsample of either morning or evening types using single-dimensional classifications (i.e., morning or evening trait type, morning or evening state type, or morning or evening trait+state type) chose the same option (either morning or evening type suggested by the LIVEMAN classification) rather than any of the six other options. The only exceptions were students from the subsample of evening types with early sleep timing (Em-types). They did not choose the evening type option suggested by the LIVEMAN classification as frequently. More frequently chosen options were lethargic and napping types (L- and N-types). Thus, the results of multidimensional classification for these Em-types contrasted with the results of classification for Me-types that were also choosing morning types from the LIVEMAN classification despite their late weekend sleep timing (Table 5).

### 2.4. Associations Between Trait, State, and Trait+State Types and Health, Mood, and Sleep Problems

The following predictions about the associations between chronotypes and health, mood, and sleep concerns can be made from the results of the comparison between single- and multidimensional classifications. Since Me-types from the single-dimensional classification of trait+state types more frequently chose morning-type rather than the other six types of LIVEMAN classification, the pattern of their associations with concerns about health, mood, and sleep can show similarity with the pattern shown by any of the morning types, including Mm-types of the single-dimensional classification of trait+state types. In contrast, the pattern of Em-types from trait+state classification cannot resemble the pattern of evening types from LIVEMAN and other classifications. This pattern is expected to be, at least partly, dissimilar to the patterns shown by other evening types, including Ee-types from the single-dimensional classification of trait+state types. Since they are more likely to choose lethargic or napping types (L- and N-types), their pattern can be more similar to the patterns of these L- and N-types of the LIVEMAN classification.

In full agreement with these expectations, the results shown in Table 6 suggest that the pattern of associations between Me-types and health, mood, and sleep concerns was very similar to the patterns of other morning types, including Mm-types of the trait+state typology. They are more likely than other types to be concerned about their sleep in the middle and end of the night but less likely to be concerned about their sleep in the beginning of the night. They were also not more concerned than other types about daytime sleepiness and depressed mood (Table 6).

Moreover, in full agreement with the expectations, the pattern of Em-types of trait+state typology differed from the patterns of other evening types (i.e., Ee-types of trait+state typology and evening types of the trait, state, and LIVEMAN typologies). The pattern of Em-types only partially resembled the patterns of other evening types, and this resemblance can be explained by partial similarity between the patterns of evening types and lethargic or napping types. For instance, Em-, Ee-, L-, and N-types suffered from daytime sleepiness and depressed mood more frequently than expected. Em-types were similar to Mm-, Me-, M-, Neither-, L-, and V-types but were dissimilar to evening (trait, state, and Ee-) types of unidimensional classifications in reporting sleep problems in the middle of the night more frequently than expected. Em-, L-, and N-types, but not Ee-types, reported problems with sleep at the beginning of the night less often than expected. Em-types were similar to Ee- and other evening types with a lower frequency of concerns about sleep at the end of the night, but, unlike any of the other evening types, Em-types did not report poorer health compared to morning types. These results support the conclusion that the partial resemblance between the patterns of Em-types and other evening types, including Ee-types of the trait+state typology, can be explained by the partial similarity of the patterns of evening types and two other (lethargic and napping) types of the LIVEMAN classification (Table 6).

Overall, the results of the comparison of patterns of associations between chronotypes and health, mood, and sleep concerns supported expectations about the patterns of Me- and Em-types of unidimensional trait+state typology (i.e., the two types with discordant trait and state assessments). Me-types were similar to the morning types of any other typology in this pattern, despite their discordant trait- and state-characteristics (i.e., late rather than early weekend sleep timing). In contrast, the pattern of Em-types with discordant trait- and state-characteristics was more similar to the patterns of chronology types other than evening types from the LIVEMAN typology (i.e., the patterns of L- and N-types).

### 2.5. An Additional Parameter in Simulations of Sleep Times Reported by Trait+State Types

Sleep times reported by five trait+state types were simulated to link the discordance between trait and state characteristics to the parameters of the underlying processes of sleep–wake regulation (Table 7). The model-based simulations of sleep–wake cycles in daytime learners/workers usually assume that bedtimes are exclusively determined by the individual’s biological clock, while risetimes on weekdays are determined by the social clocks that conflict with the biological clocks. This assumption was supported by a simulation of sleep times in three of five trait+state types: Me-, Neither-, and Em-types (Table 7).

The results of the Me- and Em-types suggested the necessity of incorporating an additional assumption in the model-based simulations of their sleep–wake cycles to account for a delay in the transition between wakefulness and sleep in these types. It seems that their bedtime cannot be exclusively determined by the biological (body) regulators of their sleep–wake cycle (as shown in Figure 2A for Mm- and Ee-types). As shown in Figure 2B, it is necessary to account for the possibility of prolongation of wakefulness beyond the bedtime set by their biological sleep–wake regulating mechanisms. Such self-deprivation of sleep in early night hours appears to reflect an additional influence of “mental (mind) clocks” on bedtimes.

In terms of the model, the previous simulations and the present simulations of sleep–wake cycles of Mm-, Neither-, and Ee-types are based on an assumption that the time of transition between wakefulness and sleep is determined by two internal (homeostatic and circadian) components of the mechanism of sleep–wake regulation. Therefore, bedtimes always coincide with *t2*, that is, the time point at which the highest buildup (*S_d_*) is reached by the process of sleep–wake regulation (*S*). The prolongation of wakefulness after this time point is not proposed for any bedtimes (either on weekdays or weekends) in the present simulations of sleep cycles reported by the types with concordant trait and state assessments (Mm- and Ee-types). The last line in Table 7 and Figure 2A shows that the model was capable of accurately simulating sleep times in these Mm- and Ee-types. However, the sleep cycles of Me- and Em-types characterized by discordant trait and state assessments cannot be accurately simulated without the additional assumption that wakefulness was prolonged after reaching *S_d_*. Their simulations required accounting for a kind of partial self-deprivation of either weekend or weekday sleep in either Me- or Em-types, respectively (Table 7 and Figure 2B). Thus, the prolongation of wakefulness after reaching the time point of the highest buildup (*S_d_*) of the process of sleep–wake regulation (*S*) was additionally proposed on weekdays in Em-types (Table 7 and Figure 2B), while such a prolongation of wakefulness in Me-types was proposed on weekdays (Table 7 and Figure 2B).

Overall, given that weekday risetimes are set by social rather than biological (internal) clocks, they occur earlier than *ad lib* weekend risetimes set by biological rather than social clocks. Consequently, the model predicts that weekday bedtimes after earlier weekday risetimes cannot occur later than weekend bedtimes after later *ad lib* weekend risetimes. If weekday bedtimes occurred later than weekend bedtimes, these bedtimes cannot be predicted and simulated by the model, postulating that these times are exclusively determined by the internal biological mechanisms of sleep–wake regulation. In order to set weekday bedtimes later than weekend bedtimes, an additional assumption was included in the model. Since weekday bedtime cannot occur exactly at *t2* in Em-types (this is the time point at which the highest buildup,${\;S}_{d}$, of the sleep–wake regulating process, S, is reached), a delay in the transition between the wake and sleep phases of the sleep–wake cycle (+h) was proposed. This proposition provided a model-based explanation for the discordant trait and state characteristics in these Em-types. Similarly, such a delay was proposed for Me-types, but on weekends rather than on weekdays. As mentioned in the introduction, students of these types might sleep late on free days because, for instance, they are trying to remain in sync with their peers who practice late sleep timing on these days. To remain active during the early night hours of these days, they can ignore the bedtime signal sent by their biological mechanisms of sleep–wake regulation (Table 7 and Figure 2B).

The results of the simulations were in agreement with the empirical results reported in Table 2, Table 3, and Table 6. Since socially determined early wake-ups reduce weekday sleep duration, the profound reduction in the time in bed on weekdays in Em-types can explain their concern about disturbances in their sleep–wake cycle, such as daytime sleepiness (Table 3). As expected, Em-types resembled L- and N-types of the LIVEMAN typology in this respect (Table 6).

We also found that the differences between five trait+state types in their simulated circadian phases (Figure 2 and Table 7) were in agreement with the differences between these types in morningness-eveningness self-assessments (e.g., Figure 1A and Table 2).

To summarize, the simulation results provided an explanation for the results of the comparison between single- and multidimensional classifications of chronotypes on the pattern of their associations with health, mood, and sleep concerns. Me-types from the single-dimensional trait+state typology most likely belonged to morning types rather than the other six types of the LIVEMAN typology. The simulations confirmed the morningness of these Me-types (an earlier rather than later circadian phase and, therefore, earlier sleep times set by their biological clocks). Consequently, the pattern of associations between these Me-types and health, mood, and sleep concerns was expected to show a similarity with the patterns shown by any other morning types, including Mm-types of the trait+state classification and morning types of the LIVEMAN classification. This expectation was supported by the study results (Table 6). In contrast, the simulations did not confirm eveningness of Em-types from the single-dimensional trait+state typology. The simulations suggested an intermediate rather than evening circadian phase and, therefore, earlier rather than later sleep times set by their biological clocks. Consequently, this simulation result explained why Em-types did not resemble evening types from the LIVEMAN classification regarding the pattern of their associations with health, mood, and sleep concerns. This pattern only partly resembled the patterns of other evening types and showed much closer similarity to the patterns of lethargic or napping types from the LIVEMAN classification (Table 6).

### 2.6. Summary of the Study Results as the Answers to Four Questions

Table 8 summarizes the answers to the questions of the present study about chronotypes with discordant trait and state characteristics of morningness-eveningness. Question #1: How many such chronotypes are there? Given that the correlations between trait and state assessments are positive in sign, chronotypes with discordant trait and state characteristics of morningness-eveningness (Me- and Em-types) were rare compared to chronotypes with concordant characteristics (Mm- and Ee-types), and the lowest rate of evening types with early weekend sleep timing (Em-types) contrasted with the highest rate of evening types with late weekend sleep timing (Ee-types). Question #2: Can multidimensional chronotyping confirm unidimensional categorization of these chronotypes as either morning or evening types? The multidimensional chronotyping did not confirm the unidimensional categorization of Em-types as evening types but did confirm the unidimensional categorization of Me-types as morning types despite their late weekend sleep timing. Question #3: Can the patterns of association between these chronotypes and health, mood, and sleep problems resemble the patterns established for chronotypes with concordant trait and state characteristics? Consequently, the pattern of associations between health, mood, and sleep problems and morning types with late weekend sleep timing (Me-types) resembled the patterns of morning types with concordant trait and state characteristics (including Mm-types), while the pattern of evening types with early weekend sleep timing (Em-types) did not resemble the patterns of evening types with concordant trait and state characteristics (including Ee-types). Question #4: Can a model of sleep–wake regulation indicate an additional regulator of sleep times underlying the discrepancy between trait and state characteristics? The model-based simulations confirmed an earlier circadian phase of morning types with late weekend sleep timing (Me-types), and this late weekend sleep timing was explained by the prolonged buildup phase of the sleep–wake regulation processes after reaching the bedtime set on weekends by these processes. In contrast, the simulations did not confirm a late circadian phase of evening types with early weekend sleep timing (Em-types); these types were more likely to have an intermediate circadian phase resembling that of afternoon and napping types, which allowed them to go to bed and wake up early on weekends.

## 3. Discussion

The questionnaire tools for trait and state assessments of morningness-eveningness were implemented in numerous studies exploring the associations between this dimension of individual chronobiological variation and health, mood, and sleep problems. The results were in favor of morning rather than evening types. In the present study, we addressed the following questions about morning and evening chronotypes with discordant trait and state characteristics of morningness-eveningness: 1. How many morning and evening types are there among university students? 2. Can the students belong to types other than morning or evening, as suggested by the multidimensional chronotyping? 3. Can these other types resemble morning and evening types with concordant trait and state characteristics regarding their pattern of associations with health, mood, and sleep concerns? 4. Can the conflicting trait and state characteristics be linked to the parameters of the underlying processes of sleep–wake regulation? We found that if trait-like assessments suggested morningness, it was supported by multidimensional chronotyping despite the atypical (late) timing of weekend sleep. In contrast, if trait-like assessments suggested eveningness, multidimensional chronotyping failed to support it. More likely, evening types who reported atypical (early) timing for weekend sleep were lethargic or napping types. Consequently, these evening types with early weekend sleep timing differed from typical evening types in health, mood, and sleep concerns. Their concerns more resembled the concerns of lethargic or napping types than the concerns of evening types. In contrast, morning types with late weekend sleep timing did not differ from other morning types in these concerns. It seems that weekend sleep mistiming did not significantly impact the health, mood, and sleep of these morning types. Simulations of sleep times reported by chronotypes with discordant trait and state characteristics of morningness-eveningness suggested that they can practice a voluntary prolongation of wakefulness to the early night hours, thus ignoring the bedtime signal from their internal biological clocks, either on weekends (morning types with late weekend sleep timing) or on weekdays (evening types with early weekend sleep timing).

This dismissal of biologically determined bedtimes revealed by the present simulations was previously addressed in the literature on the adverse effects of delayed sleep times under the terms “bedtime procrastination” [65,66,67,68,69] and “mind-bedtime procrastination” [70]. This procrastination was determined as “the volitional delay of sleep without external influences” [71]. Prior to the introduction of these terms, the concept of the conflicting social and biological clocks was proposed [72]. This concept postulates that weekday risetimes occur earlier than risetimes after *ad lib* (e.g., weekend) sleep because social clocks determine early wake-ups in contrast to later weekend risetimes set by biological clocks. If the former risetimes occur due to forced awakening, the latter risetimes occur after *ad lib* sleep. As for bedtimes, this concept postulates that they are solely determined by biological clocks, and therefore, people cannot voluntarily advance the endogenously determined weekday bedtimes to fully compensate for the reduction in sleep duration on weekdays [72]. However, this concept of two conflicting clocks does not take into account the possibility of the contribution of “mind-bedtime procrastination” to bedtimes on weekdays and/or weekends. Although sleep onset cannot be successfully shifted ahead due to the resistance of biological clocks, it can, at least sometimes, lag behind the sleep onset set by these clocks due to “the volitional delay of sleep without external influences”. The results of model-based simulations suggested that bedtimes in at least one of five trait+state subsamples cannot be explained by the model without the incorporation of an additional assumption that sleep can be partially self-deprived due to the conflict between biological clocks and the tendency of “mind-bedtime procrastination” (“mental clocks”). Overall, the simulations indicated that if the wake-ups can be determined by the conflict between social and biological clocks, bedtimes can be, at least sometimes, determined by the conflict between biological (body) and “mental (mind)” clocks that are known under the terms “mind-bedtime procrastination” and “bedtime procrastination”. When the bedtime signal sent by these biological clocks is ignored, the wake phase of the sleep–wake cycle is prolonged beyond the biologically determined bedtime, or, in other words, sleep during early night hours is self-deprived.

In agreement with the concept of three conflicting clocks, our results suggested that bedtimes of evening-type students cannot be easily shifted ahead due to the resistance of their biological clocks. Consequently, these results did not support the speculation that owl-type students can practice early weekend sleep timing, particularly in an attempt to reduce the extent of displacement of their weekday sleep relative to their internal circadian phase. Instead, the multidimensional LIVEMAN chronotyping indicated that the subsample of evening types with early weekend sleep timing mostly included lethargic and napping types rather than evening and morning types. Consequently, the patterns of association between such chronotypes and health, mood, and sleep concerns resembled the patterns of lethargic and napping types. Since our results indicated that the single-dimensional approach to the assessment of trait-like individual variation can lead to misclassification of chronotypes, the implication of multidimensional approaches to chronotyping [1,4,5,6,73,74] can be recommended to explain the conflicting results of trait and state assessments of morningness-eveningness. These study results are in agreement with studies suggesting a wider interpretation of the terms “chronotype” and “circadian type” [4,5,7,8,14,15] in studies of multidimensional individual variation in the domain of chronobiology. The SIC and other tools designed for multidimensional chronotyping can be further implemented in research regarding associations between more than two chronotypes and sleep, mood, and health problems. At least, the first attempts to implement the SIC (e.g., [75]) indicated the necessity of reconsidering the conventional view of such associations established in previous studies using the unidimensional approach to chronotyping.

We found that in the sample of university students, a third of morning types practiced late weekend sleep timing, which might be explained by the attempts of lark-type students to remain in sync with their peers who prefer late weekend sleep timing. The result suggesting that late weekend sleep timing did not change the morning type pattern of concerns about health, mood, and sleep appears to agree with the result of the comparison of these patterns before and during lockdown, which demonstrated the patterns’ persistence during lockdown [57] despite the delay in sleep timing on weekdays and weekends [76]. The previous and present results agree with the recent findings of a phenome-wide association study aimed at exploring the relationship between chronotype and 976 other phenotypes in 27,030 participants from the UK Biobank dataset. Eveningness was found to be associated with spending more time on the computer, TV, and mobile phone; playing computer games; a preference for attending pubs or social clubs; a higher frequency of consuming alcohol; active smoking; taking cannabis; tiredness and lethargy; negative emotions; depressed mood; mood swings; loneliness; and neuroticism [77]. Therefore, it can be concluded that the associations revealed in such studies seem to be internal rather than external in nature, and, consequently, a simple shift in sleep timing is unlikely to succeed in modifying these associations.

Further research can be conducted to elaborate on the potential brain mechanisms responsible for the ability to overcome the biological need for getting sleep at a certain time of the day. It seems that, unlike the circadian and sleep–wake regulating mechanisms located in the oldest brain regions, the human ability for voluntary sleep deprivation can be linked to high-level processes produced by the cerebral cortex (e.g., decision-making, problem-solving, thinking, motivation, awareness, attention, learning, etc.). Further research can also explore the integration of the concept of three conflicting—social, “mental”, and biological—clocks in a holistic approach to health developed in the theoretical framework of the biopsychosocial model proposed by Engel [78]. This model offers an improvement/alternative to the traditional biomedical model by accounting not only for biological factors but also for psychological and social factors relevant to subjective well-being and objective physical and mental health [79]. It emphasizes the importance of the biological, psychological, and social environment for understanding various conditions. In particular, one of the foundational underpinnings of this model is a revised assumption of mind/body dualism that notoriously offers a scientific account for the influence of “immaterial” mental processes on material body processes regulating human behavior [78,79].

Several limitations of this study require acknowledgement. The application of a cross-sectional and non-repeated measures design was one such limitation. In addition, we did not evaluate the numerous factors affecting sleep in university students (e.g., home environment conditions, preparation for examinations and performing deadline-sensitive homework, pre-existing health conditions and adverse health behaviors, engagement in physical activity, following a special diet, etc.). The results obtained in the study of university students cannot be generalized across age and level of education. Moreover, sleep times did not inform about the phase positions of the physiological and hormonal markers of the circadian clocks. Similarly, sleep times cannot inform about the bedtime set by the internal biological mechanisms of sleep–wake regulation. However, the lack of such information about individual differences in the circadian phase and the extent of the delay in wakefulness relative to a biologically determined bedtime cannot challenge the conclusions of our study, i.e., a need to account for the conflict of the biological and “mental” clocks to explain the discrepancies between trait and state assessments. Future studies of this conflict can focus on longitudinal evaluation of objectively measured sleep–wake cycles with parallel measurements of individual circadian phase and genetic and epigenetic markers of chronotype, as well as accounting for the contribution of multiple external factors to sleep patterns established in natural settings.

## 4. Materials and Methods

### 4.1. Survey Participants and Assessments of Chronotype, Sleep, Mood, and Health in Brief

In the winter season, lecturers invited students to respond to questions about their sleep–wake habits and behavior. The responses were collected from smartphones using Google Forms questionnaires. Informed consent was obtained from each participant in the form of the response “Agree” to the first question: “I give informed consent to anonymously and voluntarily participate in this online survey of sleep–wake behavior and habits”.

All survey participants were asked to report rise- and bedtimes on weekdays and weekends (Figure 1B and Table 3). They also completed the following questionnaires: (1) the Sleep–Wake Pattern Assessment Questionnaire (SWPAQ) for self-assessing several sleep and wake abilities (Table 2) [80], (2) the reduced (50-item) version of the Sleep–Wake Adaptability Test (SWAT) for self-assessment of abilities to sleep or wake on demand at different times of the day (Table 2) [73,81], (3) the Single-Item Chronotyping (SIC) for self-choosing a chronotype from 7 response options (Table 4 and Table 5) [6], and (4) a set of questions asking about several health, mood, and sleep concerns (Table 6) [57].

### 4.2. Details on Questionnaires for Assessment of Chronotype

The 72-item SWPAQ [80] and the 60-item SWAT [81] were constructed to assess an individual’s abilities to sleep or wake at either appropriate or inappropriate times for sleep and wakefulness. The SWPAQ includes two 12-item scales for assessing morning and evening/night components of morningness-eveningness, “Morning Lateness” (M) and “Evening Lateness” (E). The SWAT includes two 10-item scales, “Morning Sleepability” (MS) and “Nighttime Wakeability” (NW), that are the counterparts of the two SWPAQ scales (M and E, respectively). A positive/negative score on each of these scales of these questionnaires indicates an ability/inability to sleep or wake on demand in the morning (M and MS) and an ability/inability to wake or sleep on demand in the evening/at night (E and NW). The lists of items were included in the journal publications of the SWPAQ [80] and SWAT [81]. The lateness scales of the SWPAQ were previously validated against other questionnaires for trait and state assessments of morningness-eveningness [6,34,74,81,82,83,84], the objective markers of circadian phase [85,86,87,88], single-nucleotide polymorphisms in circadian clock genes [89,90], and daily rhythms of objective (e.g., spectral EEG) indexes of sleepiness [91,92].

The original English version of the SIC was proposed by O. Mairesse and A. Putilov to distinguish between six chronotypes rather than just two—morning and evening [6]. The Russian and English versions of the SIC are included in Appendix A. The English names for the 7 response options to the SIC’s question (“Self-assess your chronotype by choosing one of six patterns of daily change in alertness level”) were abbreviated as “LIVEMAN” (“Lethargic, Inconclusive, Vigilant, Evening, Morning, Afternoon, Napping” type). The illustrations and short descriptions of levels of alertness in the morning, afternoon, and evening were assigned to the graphs and names of these chronotypes (see Appendix A): (1) Lethargic type (“Moderately energetic type” in Russian): low level in the morning, afternoon, and evening; (2) Vigilant type (“Highly energetic type” in Russian): high level in the morning, afternoon, and evening; (3) Evening type: low level in the morning, moderate in the afternoon, high in the evening; (4) Morning type: high level in the morning, moderate in the afternoon, low in the evening; (5) Afternoon type (“Daytime type” in Russian): low level in the morning, high in the afternoon, moderate in the evening; (6) Napping type (“Daytime sleepy type” in Russian): high level in the morning, low in the afternoon, moderate in the evening; and (7) Inconclusive type (“None of the above”). The responses were used to calculate the ratio between the real count of each chronotype and its expected count in each of the subsamples of the whole sample (Table 4, Table 5 and Table 6).

All chronobiological questionnaires applied in the present study were cross-validated and validated against daily sleepiness curves obtained with the 19-time point Visuo-verbal Judgment Task (VJT) in several previously published studies [6,34,74,81,82].

### 4.3. Division of the Whole Sample into Subsamples and Statistical Methods

In order to distinguish between morning and evening types with concordant and discordant trait and state characteristics, the whole sample was initially subdivided into 3 subsamples according to the scores on the morningness-eveningness scales and weekend sleep times (trait and state types, respectively). The dual criteria applied when dividing the whole sample into three trait and three state subsamples are shown in the upper part of Table 1. The subdivisions based on scores and times were combined to create 5 trait+state types. The combined criteria applied when dividing the whole sample into these 5 trait+state subsamples are shown in the lower part of Table 1. Additionally, the whole sample was divided into 7 subsamples according to the responses from the SIC question (i.e., LIVEMAN types).

The SPSS_26.0_ statistical software package (IBM, Armonk, NY, USA) was used for testing the significance of differences between the subdivisions of the whole sample according to questionnaire scores and sleep times (Table 2 and Table 3, respectively). Two-way ANOVAs with the additional independent factor “Sex” (male and female university students) were run to test the significance of the main effects of the factor “Subsample” (Table 2 and Table 3, bottom line). *Post hoc* pairwise Bonferroni comparisons were used to confirm the significance of the differences between the trait+state subsamples (Table 2 and Table 3). These differences are illustrated in Figure 1. A Pearson chi-square test (χ^2^) was applied to examine the significance of the deviation between the real counts and expected counts in subsamples of three trait, three state, five trait+state, and seven LIVEMAN types (Table 6).

### 4.4. Model-Based Simulations

For the present simulations, one of the variants [93] of the two-process model of sleep–wake regulation [94] was applied. It proposes that any parameter of the homeostatic process described in the classical variant of the two-process model [94] is modulated by the circadian clocks. This modulation [93] is the counterpart of the circadian process of the classical variant of the model [94]. To maintain sleep–wake homeostasis, the homeostatic process regulates daily sleep amount/depth through the waxing (during waking) and waning (during sleep) of sleep pressure. The modulation of the homeostatic process by the circadian clocks was incorporated in the model in the form of the simplest periodic (sine) function with a circadian period. If t1 and t2 are the initial times for the buildup and decay phases of this process of sleep–wake regulation $S\left(t\right)$, it can be described as the following:(1)$$S\left(t\right)=\left[S_{u}+C\left(t\right)\right]-\left\{\left[S_{u}+C\left(t\right)\right]-S_{b}\;\right\}*e^{-\frac{t-t1}{\left[Tb-k\;*\;C\left(t\right)\right]}}\;S\left(t\right)=[S_{l}+C(t)]-\{S_{d}-[S_{l}+C(t)]\}*e^{-\frac{t-t2}{\left[Td-k\;*\;C\left(t\right)\right]}}$$
where(2)$$C(t)=A*sin(2\pi *t/\tau +\varphi_{0})$$
is a sine function with a circadian period *τ*. In the simulations, *τ* was assigned to 24.0 h because, in natural settings, the circadian clocks always remain under control of (i.e., are entrained to) the external light–dark cycle with a 24.0 h period [64,76,93].

The parameters of the model [93] were initially derived from data on (1) the durations of recovery sleep after 6 gradually increasing intervals of extended wakefulness [95] and (2) the levels of SWA in 10 naps [96] and two recovery sleep episodes [97,98]. The simulation of such experimental data provided the possibility of measuring $S\left(t\right)$ as relative slow-wave activity (rSWA). The initial parameters of the model were slightly modified in the previous (e.g., [64,76]) and present simulations (Figure 2 and Table 7) to account for the difference between the initially simulated experimental sleep durations [93,95] and times in bed calculated from self-reported bed- and risetimes on weekdays and weekends [64,76].

The present simulations (Figure 2 and Table 7) were performed using the least-squares method. In the process of such simulations, the empirical and simulated sleep times and the empirical and simulated weekday-weekend gaps in sleep times (9 time estimates in total) were compared to reach the reduction in the maximum of 9 deviations of a simulated value from an empirical value to 0.1 h (6 min) or lower (see the bottom line and the notes for Table 7). It is of importance to emphasize the difference between previous [64,76,93] and present simulations when adding an assumption that bedtimes, at least in some of the 5 subsamples of the whole sample (Figure 2B), can be delayed due to missing the bedtime signal from biological clocks, i.e., “mind-bedtime procrastination”. This leads to a prolonged wake phase of the sleep–wake cycle beyond the biologically determined bedtime, or, in other terms, to sleep self-deprivation during early night hours (Table 7 and Figure 2B).

Generative artificial intelligence (GenAI) was not used in this paper.

## 5. Conclusions

In the present study of 1582 university students, multidimensional chronotyping was applied to support the results of single-dimensional chronotyping, and the associations between discordant trait and state assessments of morningness-eveningness and health, mood, and sleep concerns were evaluated. Multidimensional chronotyping confirmed the morningness of morning types with late weekend sleep timing. Consequently, the morning-type pattern of associations with health, mood, and sleep concerns was also confirmed for these morning types with late weekend sleep timing. Such results agree with the previously published results, indicating that late sleep timing cannot modify the pattern of associations between chronotype and health, mood, and sleep problems. Multidimensional chronotyping did not support the eveningness of evening types with early weekend sleep timing. These types mostly belonged to the lethargic and napping types. Consequently, their pattern of associations with health, mood, and sleep problems was less similar to those of evening types with concordant trait and state assessments and more similar to those of lethargic and napping types. Therefore, the implication of multidimensional approaches to the classification of chronotypes can be recommended for clarifying the causes of disagreement between trait and state assessments of morningness-eveningness. The model-based simulations of sleep–wake cycles in daytime learners/workers usually assume that bedtimes, unlike risetimes on weekdays, are exclusively determined by biological clocks, while risetimes on weekdays result from the conflict between biological and social clocks. However, the present results of model-based simulations of the sleep–wake cycles of students from the subsamples with discordant trait and state assessments of morningness-eveningness indicated a more complex conflict between three rather than two kinds of time regulators for sleep. The additional regulator (“mind clocks”) was previously conceptualized under the terms “bedtime procrastination” and “mind-bedtime procrastination”. The simulations suggested that if, in accordance with the concept of two conflicting clocks, social and biological, these clocks mostly determine weekday and weekend risetime, respectively, this concept fails to recognize the complexity of clocks governing bedtimes. Biological clocks cannot exclusively determine bedtimes on weekdays and weekends in chronotypes with discordant trait and state assessments of morningness-eveningness. The simulations of their sleep–wake cycles indicated that these bedtimes were more likely determined by an additional conflict of biological clocks with “mental clocks”, previously described as “mind-bedtime procrastination” or “bedtime procrastination”. The dismissal of the bedtime signal sent by biological clocks was proposed to occur in at least the subsamples with conflicting trait and state assessments of morningness-eveningness. As a result, sleep was self-deprived during early night hours (i.e., the wake phase of the sleep–wake cycle was prolonged beyond the biologically determined bedtime).

## Figures and Tables

**Figure 2 clockssleep-07-00035-f002:**
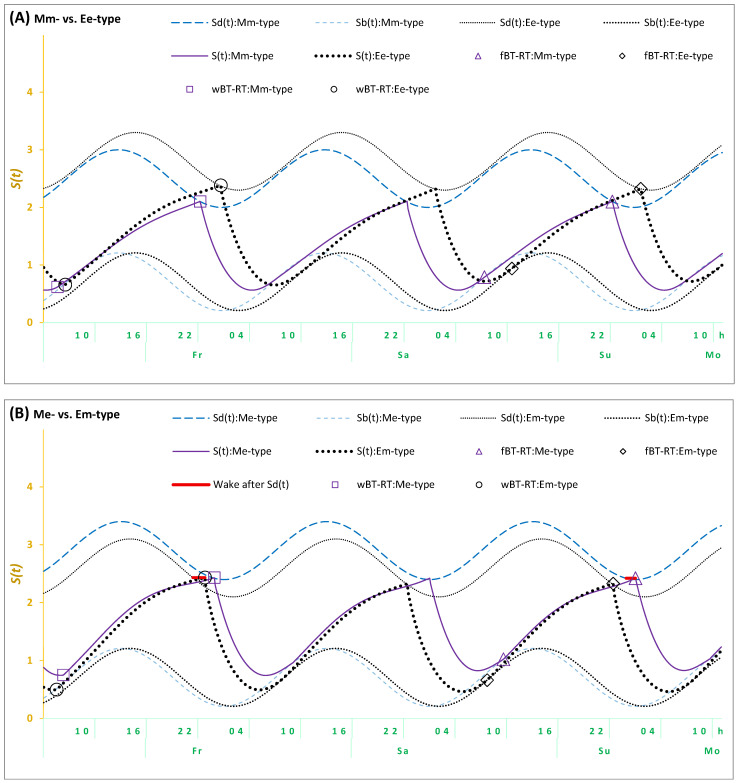
Simulations of three sleep–wake cycles in students from four trait+state types. Simulations of a sequence of three sleep–wake cycles for Fr, Sa, and Su (Friday, Saturday, and Sunday); fBT-RT: Bed- and risetimes on weekends; wBT-RT: Bed- and risetimes on weekdays. Mm-, Me-, Em-, and Ee-types: Distribution into trait+state subsamples according to the combinations of scores on morningness-eveningness scales and weekend sleep times. ${\;S}_{b}(t)$: Risetime set by the biological sleep–wake regulating mechanisms after *ad lib* sleep on weekends; ${\;S}_{d}(t)$: Bedtime set by these mechanisms on weekends and weekdays. (**A**) Mm-type vs. Ee-type, (**B**) Me-type vs. Em-type. Wake after ${\;S}_{d}(t)$: The **solid red line** illustrates a delay of either weekend or weekday bedtime in either Me- or Em-types, respectively, after reaching ${\;S}_{d}(t)$. This delay was added in the simulations of the sleep–wake cycles in (**B**), but not in (**A**). See the parameters of the model in Table 7.

**Table 1 clockssleep-07-00035-t001:** Assignment of students to three trait types, three state types, and five trait+state types.

Abbreviation	Assessment	Criterion	Trait or State Type			Count
Distribution of students into three—either trait or state—types						
	Sum of	Trait type	M-	Sum	E-	Total
Trait	scores on	Dual	≤0	M+E	>0	
type	two scales	criterion	≤0	MS+NW	>0	
M-	M+E	≤0	423	197	0	620
E-	M+E	>0	0	195	767	962
	Total	Trait type	423	392	767	1582
M-	MS+NW	≤0	423	195	0	618
E-	MS+NW	>0	0	197	767	964
	Total	Trait type	423	392	767	1582
	Weekend bed-	State type	m-	Weekend	e-	Total
State	or risetime,	Dual	≤12 a.m.	bedtime	>12 a.m.	
type	clock h	criterion	≤9 a.m.	risetime	>9 a.m.	
m-	Bedtime	≤12 a.m.	321	252	0	573
e-	Bedtime	>12 a.m.	0	257	752	1009
	Total	State type	321	509	752	1582
m-	Risetime	≤9 a.m.	321	257	0	578
e-	Risetime	>9 a.m.	0	252	752	1004
	Total	State type	321	509	752	1582
Distribution of students into five trait+state types by combining trait types with state types						
Trait+	Weekend bed-	Trait type	M-	Sum	E-	Total
state	and risetime,	Dual	≤0	M+E	>0	Trait+state
type	clock h	criteria	≤0	MS+NW	>0	type
Mm-	Bedtime, risetime	≤12 a.m., ≤9 a.m.	183	0	0	183
Me-	Bedtime, risetime	>12 a.m., >9 a.m.	86	0	0	**86**
Neither-	Bedtime, risetime	Other	154	392	214	760
Em-	Bedtime, risetime	≤12 a.m., ≤9 a.m.	0	0	55	**55**
Ee-	Bedtime, risetime	>12 a.m., >9 a.m.	0	0	498	498
	Total	Trait type	423	392	767	1582
Trait+	Sums of	State type	m-	Weekend	e-	Total
state	scores on	Dual	≤12 a.m.	bedtime	>12 a.m.	Trait+state
type	two scales	criteria	≤9 a.m.	risetime	>9 a.m.	type
Mm-	M+E, MS+NW	≤0,≤0	183	0	0	183
Me-	M+E, MS+NW	>0,>0	0	0	86	**86**
Neither-	M+E, MS+NW	Neither	83	509	168	760
Em-	M+E, MS+NW	≤0,≤0	55	0	0	**55**
Ee-	M+E, MS+NW	>0,>0	0	0	498	498
	Total	State type	321	509	752	1582

**Notes**: The university students (n = 1582) were distributed into subsamples according to their trait, state, or trait+state type using scores on morningness-eveningness scales, weekend sleep timing, or a combination, respectively. M+E and MS+NW: Division of trait subsamples according to the sums of scores on scales designed to assess morning and evening/night components of morningness-eveningness (12-item M and E scales of the SWPAQ named “Morning Lateness” and “Evening Lateness”, respectively, and 10-item MS and NW scales of the SWAT named “Morning Sleepability” and “Nighttime Wakeability”, respectively). Each of these four scales consists of an equal number of positively and negatively worded statements (six and six in the SWPAQ and five and five in the SWAT). Half of the statements are positively worded, while the other half are negatively worded (i.e., they describe the habits or behaviors in a way that highlights either eveningness or morningness, respectively). Therefore, the responses to these statements are assigned as either +1 or −1 to indicate eveningness or morningness, respectively. Consequently, the sums of scores on two morningness-eveningness scales of each questionnaire (M+E and MS+NW) allowed the subdivision of the whole sample into subsamples of morning and evening types (M- and E-types, respectively), and, when the subdivisions based on scales of two questionnaires were combined (i.e., Dual criterion), this provided the subdivision into three trait subsamples (M-type with M+E and MS+NW ≤ 0, other type, and E-type with M+E and MS+NW > 0). Similarly, the whole sample was subdivided according to weekend bed- and risetimes not later or later than 12 a.m. and 9 a.m. (approximately one-third and two-thirds of students, respectively). These subdivisions, according to bed- and risetimes, provided the possibility of subdividing the whole sample into three state subsamples (m-type with bedtime not later than 12 a.m. and risetime not later than 9 a.m., other type, and e-type with bedtime later than 12 a.m. and risetime later than 9 a.m.). The combination of subdivisions into three (trait and state) subsamples gave the final subdivision of the whole sample into five subsamples: (1) Mm-type with M+E and MS+NW ≤ 0, bedtime not later than 12 a.m., and risetime not later than 9 a.m. (concordant morning-type assessments); (2) Me-type with M+E and MS+NW ≤ 0, but bedtime later than 12 a.m. and risetime later than 9 a.m. (discordant morning-type assessments); (3) Neither-type; (4) Em-type with M+E and MS+NW > 0, but with bedtime not later than 12 a.m. and risetime not later than 9 a.m. (discordant evening-type assessments); and (5) Ee-type with M+E and MS+NW > 0, bedtime later than 12 a.m. and risetime later than 9 a.m. (concordant evening-type assessments). The types with concordant assessments of morningness-eveningness (Mm- and Ee-types) were separated from the types with discordant trait and state assessments (Me- and Em-types) by such a subdivision into five trait+state types. Printed in **bold**: Number of university students assigned to the subsamples with discordant trait and state assessments (Me- and Em-types).

**Table 2 clockssleep-07-00035-t002:** F-ratios and *post hoc* pairwise comparisons of questionnaire scores in five trait+state types.

Questionnaire		SWPAQ		SWAT		SWPAQ	SWAT
	Scale	M	E	MS	NW	M+E	MS+NW
Mm- vs.	Me-	−1.9 *	−1.6	−2.2 **	−0.9	−3.5 **	−3.0 ***
	Neither-	−5.0 ***	−5.1 ***	−5.0 ***	−3.2 ***	−10.1 ***	−8.2 ***
	Em-	−8.3 ***	−7.7 ***	−8.0 ***	−4.8 ***	−16.0 ***	−12.8 ***
	Ee-	−9.9 ***	−9.7 ***	−9.0 ***	−6.5 ***	−19.5 ***	−15.5 ***
Me- vs.	Neither-	−3.1 ***	−3.5 ***	−2.8 ***	−2.3 ***	−6.6 ***	−5.1 ***
	Em-	−6.4 ***	−6.1 ***	−5.8 ***	−4.0 ***	−12.5 ***	−9.7 ***
	Ee-	−8.0 ***	−8.1 ***	−6.8 ***	−5.6 ***	−16.0 ***	−12.4 ***
Neither- vs.	Em-	−3.3 ***	−2.6 **	−3.0 ***	−1.6 *	−5.9 ***	−4.6 ***
	Ee-	−4.9 ***	−4.6 ***	−4.0 ***	−3.3 ***	−9.4 ***	−7.3 ***
Em- vs.	Ee-	−1.6	−1.9	−1.0	−1.7 *	−3.5 **	−2.7 **
F_4/1752_		141.1 ***	98.4 ***	139.6 ***	107.4 ***	256.7 ***	290.1 ***

**Notes**: Differences in questionnaire scores between university students distributed into five subsamples of trait+state types using the combinations of scores on morningness-eveningness scales with weekend sleep times. F_4/1752_: F-ratio for the main effect of the independent factor “Subsample” (Mm-, Me-, Neither-, Em-, and Ee-types) from two-way ANOVAs with the additional independent factor “Sex” (male and female students). Level of significance for either F-ratio (bottom line) or *t*-test with Bonferroni correction for the number of *post hoc* pairwise comparisons: * *p* < 0.05, ** *p* < 0.01, *** *p* < 0.001. M, E, MS, and NW: Scores on scales for the assessment of morning and evening/night components of morningness-eveningness (two 12-item scales of the SWPAQ, “Morning Lateness” and “Evening Lateness”, and two 10-item scales of the SWAT, “Morning Sleepability” and “Nighttime Wakeability”); M+E and MS+NW: Sums of scores on two scales of the SWPAQ and SWAT, respectively. See Table 1 for other notes and Figure 1A for an illustration of scores in five subsamples.

**Table 3 clockssleep-07-00035-t003:** F-ratios and *post hoc* pairwise comparisons of sleep times in five trait+state types.

Sleep Time, Clock h		Weekday		Weekend		Time in Bed	
Type vs.	Type	Bedtime	Risetime	Bedtime	Risetime	Weekday	Weekend
Mm- vs.	Me-	−1.4 ***	−0.7 ***	−2.5 ***	−2.4 ***	0.7 **	0.1
	Neither-	−1.3 ***	−0.5 ***	−1.6 ***	−1.7 ***	0.8 ***	−0.1
	Em-	−0.7 **	0.1	0.1	−0.6	0.8 *	−0.6
	Ee-	−2.2 ***	−0.8 ***	−3.2 ***	−3.2 ***	1.3 ***	0.0
Me- vs.	Neither-	0.1	0.2	1.0 ***	0.7 ***	0.0	−0.3
	Em-	0.7 *	0.8 **	2.6 ***	1.8 ***	0.1	−0.8
	Ee-	−0.8 ***	−0.2	−0.7 **	−0.8 ***	0.6 *	−0.1
Neither- vs.	Em-	0.6 *	0.6 **	1.6 ***	1.1 ***	0.0	−0.5
	Ee-	−0.9 ***	−0.4 ***	−1.6 ***	−1.5 ***	0.5 ***	0.2
Em- vs.	Ee-	−1.4 ***	−0.9 ***	−3.2 ***	−2.6 ***	0.5	0.6
F_4/1752_		81.8 ***	20.8 ***	153.7 ***	177.0 ***	17.9 ***	0.9

**Notes**: Differences in sleep times between university students distributed into five subsamples of trait+state types using the combinations of scores on morningness-eveningness scales with weekend sleep times. F_4/1752_: F-ratio for the main effect of the independent factor “Subsample” (Mm-, Me-, Neither-, Em-, and Ee-types) from two-way ANOVAs with the additional independent factor “Sex” (male and female students). Level of significance for either F-ratio (bottom line) or *t*-test with Bonferroni correction for the number of *post hoc* pairwise comparisons: * *p* < 0.05, ** *p* < 0.01, *** *p* < 0.001. See other notes in Table 1 and Table 2, and see Figure 1B for an illustration of bed- and risetimes.

**Table 4 clockssleep-07-00035-t004:** Ratio between the expected and real count of LIVEMAN types among trait and state types.

Trait or State Type	Sum of Scores on Two Scales			Clock Time for Weekend		
Dual Criteria	≤0	M+E	>0	≤12 a.m.	Bedtime	>12 a.m.
LIVEMAN Type	≤0	MS+NW	>0	≤9 a.m.	Risetime	>9 a.m.
Lethargic	0.98	**1.11**	0.96	**1.21**	**1.08**	0.85
Inconclusive	0.98	0.91	**1.06**	**1.23**	0.93	0.95
Vigilant	**1.18**	**1.39**	0.70	**1.15**	**1.03**	0.92
Evening	0.25	0.74	**1.55**	0.42	0.86	**1.34**
Morning	**2.36**	**1.03**	0.23	**1.92**	**1.10**	0.54
Afternoon	0.97	**1.26**	0.88	0.96	**1.10**	0.95
Napping	0.85	0.92	**1.12**	0.93	**1.04**	1.00
Count (1582 in total)	423	392	767	321	509	752

**Notes**: Each cell contains a ratio between the real and expected number of university students. The students were distributed into three subsamples according to their sleep–wake questionnaire scores or weekend sleep timing (either trait or state types). Scores on scales of two sleep–wake questionnaires were used to divide students into trait types (left), and weekend sleep timing was used to divide students into state types (right). M+E and MS+NW: Division into subsamples according to the sums of scores on scales assessing morning and evening/night components of morningness-eveningness (two 12-item M and E scales of the SWPAQ, “Morning Lateness” and “Evening Lateness”, and two 10-item MS and NW scales of the SWAT, “Morning Sleepability” and “Nighttime Wakeability”). The dual criterion for the division into three subsamples (left) was either a negative or positive sum of M and E scores combined with either a negative or positive sum of MS and NW scores. The dual criterion for the division into three subsamples according to weekend sleep timing (right) was bed- and risetime on weekends either not later or later than 12 a.m. and 9 a.m., respectively. LIVEMAN type: Students were additionally distributed into seven LIVEMAN types (lines). These types were chosen among seven options illustrating and shortly describing the patterns of daily change in alertness level: L(ethargic): low in the morning, afternoon, and evening; I(nconclusive): none of the six other options; V(igilant): high in the morning, afternoon, and evening; E(vening): low in the morning, moderate in the afternoon, and high in the evening; M(orning): high in the morning, moderate in the afternoon, and low in the evening; A(fternoon): low in the morning, high in the afternoon, and moderate in the evening; N(apping): high in the morning, low in the afternoon, and moderate in the evening (see also Appendix A). Printed in **bold**: The real count is greater than the expected count. The level of significance (*p*) for the results of the two-sided Pearson Chi-Square test was always < 0.001.

**Table 5 clockssleep-07-00035-t005:** Ratio between the expected and real count of LIVEMAN types among trait+state types.

Trait+State Type	Mm-	Me-	Neither-	Em-	Ee-	Count
Lethargic	**1.25**	**1.05**	**1.06**	**1.67**	0.74	69
Inconclusive	**1.61**	0.23	0.99	**1.07**	0.91	80
Vigilant	**1.10**	**1.11**	**1.14**	**1.09**	0.72	133
Evening	0.12	0.50	0.83	0.95	**1.67**	515
Morning	**2.57**	**1.95**	**1.07**	0.78	0.17	293
Afternoon	0.83	**1.21**	**1.15**	0.86	0.82	334
Napping	0.82	0.70	0.96	**1.45**	**1.13**	158
Count	183	86	760	55	498	1582

**Notes**: Each cell contains a ratio between the real and expected number of university students. The students were distributed into five subsamples according to combinations of their sleep–wake questionnaire scores and weekend sleep timing (trait+state types). Printed in **bold**: The real count is greater than the expected count. The level of significance (*p*) for the results of the two-sided Pearson chi-square test was always < 0.001. See Table 1, Table 2, Table 3 and Table 4 for other notes.

**Table 6 clockssleep-07-00035-t006:** Associations between chronotypes and concerns about health, mood, and sleep.

Type	Trait: Sum of Scores on Two Scales				State: Clock Time for Weekend			
Dual Criteria	≤0	M+E	>0		≤12 a.m.	Bedtime	>12 a.m.	
Concern About	≤0	MS+NW	>0	χ^2^_2_	≤9 a.m.	Risetime	>9 a.m.	χ^2^_2_
Not good health	0.81	0.90	**1.16**	29.4 ***	0.83	0.96	**1.10**	13.3 **
Depressed mood	0.69	0.90	**1.22**	14.5 **	0.77	0.87	**1.18**	9.0 *
Evening sleep	0.60	1.00	**1.22**	24.0 ***	0.66	0.77	**1.30**	30.3 ***
Nighttime sleep	**1.55**	**1.03**	0.69	8.3 *	**1.33**	**1.08**	0.80	2.8
Morning sleep	**1.92**	**1.08**	0.45	42.2 ***	**1.60**	0.98	0.76	11.2 **
Daytime sleepiness	0.58	0.85	**1.31**	53.3 ***	0.83	**1.02**	**1.06**	4.4
Any of these four	0.82	0.94	**1.13**	33.5 ***	0.90	0.93	**1.09**	13.8 **
Partner’s sleep	0.85	0.99	**1.09**	12.6 **	0.93	0.91	**1.09**	10.2 **
Count (1582 in sum)	423	392	767		321	509	752	
**Trait+State Type**	**Mm-**	**Me-**	**Neither-**	**Em-**	**Ee-**	**χ^2^_4_**	**Yes Count**	**No Count**
Not good health	0.70	**1.03**	0.96	1.00	**1.17**	25.0 ***	694	888
Depressed mood	0.62	0.78	0.94	**1.08**	**1.27**	12.5 *	238	1344
Evening sleep	0.55	0.86	0.87	0.77	**1.41**	31.6 ***	298	1284
Nighttime sleep	**1.51**	**1.76**	**1.12**	**1.36**	0.45	10.9 *	63	1519
Morning sleep	**2.15**	**1.75**	**1.03**	0.54	0.45	31.8 ***	105	1477
Daytime sleepiness	0.52	0.65	0.95	**1.80**	**1.23**	39.6 ***	400	1182
Any of these four	0.80	0.93	0.94	**1.26**	**1.14**	28.8 ***	866	716
Partner’s sleep	0.92	0.89	0.96	0.86	**1.13**	10.0 *	706	876
Count (1582 in sum)	183	86	760	55	498			
**LIVEMAN Type**	**L-**	**I-**	**V-**	**E-**	**M-**	**A-**	**N-**	**χ^2^_6_**
Not good health	**1.49**	0.83	0.65	**1.12**	0.75	**1.02**	**1.23**	53.7 ***
Depressed mood	**2.60**	1.00	0.20	0.98	0.77	**1.14**	**1.18**	51.1 ***
Evening sleep	0.85	0.86	0.76	**1.26**	0.98	0.89	0.77	13.3 *
Nighttime sleep	**1.48**	0.63	**1.13**	0.98	**1.11**	0.90	0.95	1.5
Morning sleep	**1.96**	0.57	**1.59**	0.73	**1.44**	0.77	0.86	17.1 **
Daytime sleepiness	**1.44**	0.59	0.12	**1.25**	0.61	0.91	**1.85**	109.8 ***
Any of these four	**1.30**	0.68	0.59	**1.17**	0.87	0.89	**1.29**	89.7 ***
Partner’s sleep	**1.23**	0.95	0.96	**1.11**	0.88	0.89	**1.06**	15.8 *
Count (1582 in sum)	69	80	133	515	293	334	158	

**Notes**: Ratio between the expected and real count of university students distributed into subsamples according to trait, state, trait+state, or LIVEMAN typology. Printed in **bold**: The real count is greater than the expected count; χ^2^_2_ or χ^2^_4_ or χ^2^_6_: Level of significance of the two-sided Pearson chi-square test (* *p* < 0.05, ** *p* < 0.01, *** *p* < 0.001). Health was dichotomized as Not good (either Intermediate or Bad) and Good; Mood was dichotomized as Depressed and any other (Normal or Euphoric or Other); Evening sleep, Nighttime sleep, Morning sleep, and Daytime sleepiness: The choice among four sleep concerns (difficulties with falling asleep, awakenings during the night, too early morning awakening, and excessive daytime sleepiness). Any of these four: Choosing one of these four options in response to the question, “What is your main concern about your sleep?” Partner’s sleep: The response “Yes” to the question, “Do you have a partner complaining about your sleep or your sleep-related daytime performance?” L-, I-, V-, E-, M-, A-, N-: LIVEMAN types were chosen from seven response options (see Appendix A). See other notes for Table 1, Table 2, Table 3, Table 4 and Table 5.

**Table 7 clockssleep-07-00035-t007:** Parameters of the model of sleep–wake regulation in five trait+state types.

Trait+State Type		Mm-	Me-	Neither-	Em-	Ee-
Inverse exponential buildup (1a) and exponential decay phases (1b) of the sleep–wake regulating process *S*(*t*)	*S_b_* (lowest decay), rSWA	0.70	0.70	0.70	0.70	0.70
	*S_d_* (highest buildup), rSWA	2.50	2.90	2.60	2.60	2.80
	*S_l_* (lower asymptote), rSWA	0.70	0.70	0.70	0.70	0.70
	*S_u_* (upper asymptote), rSWA	7.80	5.00	6.50	6.00	6.90
	*T_d_* (time constant for decay), h	66.38	28.88	49.07	36.15	50.59
	*T_b_* (time constant for buildup), h	1.97	2.37	2.03	1.69	2.24
24-h sine wave-form modulation (2) of parameters of the buildup (1a) and decay phases (1b) of *S*(*t*)	*φ_max_* (circadian peak), clock h	13.69	14.03	15.10	15.06	15.68
	*A* (circadian amplitude), rSWA	0.50	0.50	0.50	0.50	0.50
	*τ* (circadian period), h	24.00	24.00	24.00	24.00	24.00
	*k* (doubled circadian impact)	2.00	2.00	2.00	2.00	2.00
Initial times for the buildup (1a) and decay phases (1b) of *S*(*t*)	*t1*, clock h, weekend risetime	8.23	10.58	9.90	8.63	11.4
	*t2*, clock h, weekend bedtime	23.26	1.96	24.94	23.33	2.53
Prolongation of the buildup	+ h on weekday	0.00	0.00	0.00	**1.00**	0.00
phase after reaching *t2*	+ h on weekend	0.00	**0.30**	0.00	0.00	0.00
Maximal of nine deviations of simulated sleep time from reported sleep time, h		0.06	0.04	0.03	0.02	0.04

**Notes**: Mm-, Me-, Neither-, Em-, and Ee-types: The distribution of students into five trait+state types. The difference between the present and previously published simulations resulted from accounting for the possibility of sleep self-deprivation by prolongation of the buildup phase behind t2 (+h) on either weekends or weekdays in either Me- or Em-types, respectively. Printed in **bold**: +h is larger than zero. To highlight the difference between the subsamples in the circadian phase (*φ_max_*), the variation in some of the model parameters was not allowed. A maximum of nine deviations of simulated sleep time from reported sleep time were recorded. h: a maximum of nine differences between the simulated and empirical estimates (time in bed and bed- and risetimes on weekdays and weekends, and weekend–weekday gaps in these sleep times). Clock time is given in decimal hours. See Figure 2B and Figure 2A for an illustration of the simulations with and without prolongation of the buildup phase behind t2, respectively.

**Table 8 clockssleep-07-00035-t008:** Answers to the study questions #1–#4.

	Trait+State Type	Mm-	Me-	Em-	Ee-
Unidimensional	Unidimensional trait chronotyping				
chronotyping	Morning type trait	+	+		
	Evening type trait			+	+
	Unidimensional state chronotyping				
	Morning type state	+		+	
	Evening type state		+		+
Question #1	1. Number and percentage of students, 1582 in total (Table 1)				
	Count	183	86	55	498
	Percentage, %	11.57	5.44	3.48	31.48
Question #2	2. Confirmation: multidimensional chronotyping (Table 5)				
	Morning (M-) type	+	+		
	Evening (E-) type				+
	Other (L- or N-) types			+	
Question #3	3. Confirmation: associations with health, mood, and sleep concerns (Table 6)				
	Morning (M-) type	+	+		
	Evening (E-) type				+
	Other (L- or N-) types			+	
Question #4	4a. Confirmation: position of the circadian peak (Table 7)				
	Early phase (Morning type)	+	+		
	Late phase (Evening type)				+
	Other phase (Intermediate type)			+	
	4b. Confirmation: prolonged buildup (wake) phase after reaching *t2*				
	Without prolongation	+			+
	Prolongation on weekends		+		
	Prolongation on weekdays			+	

**Notes**: Upper part of the table: Results of unidimensional trait, state, and trait+state chronotyping. Question #1: Results of the count and percentage of university students distributed into subsamples according to trait+state typology. Of 1582 students, 8.9% (Me- and Em-types) reported discordant trait and state assessments of morningness-eveningness. Question #2: Results of the confirmation of trait+state typology by LIVEMAN typology. Morning type was confirmed for Me-types, while evening type was not confirmed for Em-types. Question #3: Results of the confirmation of the associations for trait+state typology by the associations for LIVEMAN typology. For Me-types, the associations for other morning types were confirmed, while for Em-types, the associations for other evening types were not confirmed. Question #4: Results of the confirmation of circadian phase (a) and the necessity of accounting for bedtime delays (b). For Me-types, the early circadian phase (4a) was confirmed despite the weekend sleep delay due to the prolonged buildup phase of the sleep–wake regulation processes after reaching *t2*, that is, the bedtime set by these processes on weekends (4b). For Em-types, the late circadian phase (4a) was not confirmed despite the weekday sleep delay due to the prolonged buildup phase of the sleep–wake regulation processes after reaching *t2* (4b). See other notes for Table 1, Table 5, Table 6 and Table 7.

## Data Availability

Data from the survey and simulation results are available from the first author on reasonable request.

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
