# Peer review of "The Owls Are Not What They Seem: Health, Mood, and Sleep Problems Reported by Morning and Evening Types with Atypical Timing of Weekend Sleep"

_2624-5175, 2025, doi:10.3390/clockssleep7030035_

Round 1
Reviewer 1 Report
Comments and Suggestions for Authors
The scope and aims of the submission remain vage/unclear. For example, it is not clear what the authors mean with "challenge" in their abstract.
If the MEQ or MCTQ are to be challenged, then the new tool (LIVEMAN) should be validated against the established tools and questionnaires.
There are a number of papers already dealing with this subject and should be cited and discussed accordingly:
# PMID: 25923181
# PMID: 26243627
# PMID: 34204181
I suggest the authors to restructure their manuscript to be more precise in its scope. Also, to cite the relevant papers. The idea to further develop the concept of chronotypology is timely, but the current submission requires some more red line and up-to-date citations.
Comments on the Quality of English Language
Need to be improved substantially.
Author Response
|
Response to Reviewer #1 Comments
|
||
|
1. Summary |
|
|
|
Please find the detailed responses below and the corresponding revisions/corrections highlighted in the re-submitted files in red.
|
||
|
2. Questions for General Evaluation |
Reviewer’s Evaluation |
Response and Revisions |
|
Does the introduction provide sufficient background and include all relevant references? |
Must be improved/Not applicable |
The introduction section was rewritten |
|
Are all the cited references relevant to the research? |
Can be improved/Must be improved/Not applicable |
The list of references was doubled |
|
Is the research design appropriate? |
Can be improved/Must be improved/Not applicable |
The result section was revised |
|
Are the methods adequately described? |
Must be improved/Not applicable |
The method section was extended and revised |
|
Are the results clearly presented? |
Must be improved/Not applicable |
The result section was revised and, in particular, restructured to introduce more details on the methodology before the method section |
|
Are the conclusions supported by the results? |
Can be improved/Must be improved/Not applicable |
The conclusion section was revised |
|
3. Point-by-point response to Comments and Suggestions for Authors |
||
|
Comments 1: The scope and aims of the submission remain vage/unclear. For example, it is not clear what the authors mean with "challenge" in their abstract. |
||
|
Response 1: Thank you for pointing this out. We agree with this comment. Abstract was rewritten to make its content clearer. |
||
|
Comments 2: If the MEQ or MCTQ are to be challenged, then the new tool (LIVEMAN) should be validated against the established tools and questionnaires. |
||
|
Response 2: More details were provided for the SIC, both in the results and methods sections, we also added the references to several publications were LIVEMAN typology was validated against several other questionnaires and the daily sleepiness curves of the VJT. Comments 3: There are a number of papers already dealing with this subject and should be cited and discussed accordingly: # PMID: 25923181 # PMID: 26243627 # PMID: 34204181. Response 3: Thank you for pointing this out. We agree with this comment. The list of references was doubled, and these particular publications were cited in a new paragraph of Introduction. Comments 4: I suggest the authors to restructure their manuscript to be more precise in its scope. Also, to cite the relevant papers. The idea to further develop the concept of chronotypology is timely, but the current submission requires some more red line and up-to-date citations. Response 4: See the previous reply, and the manuscript was restructured to include for references and results on the issue of implication of multidimensional approach to chronotyping. Thank you very much indeed for these recommendations that helped to revise and improve the entire manuscript. |
||
|
4. Response to Comments on the Quality of English Language |
||
|
Point 1: Need to be improved substantially |
||
|
Response 1: English was improved everywhere in the manuscript, and the revised version was checked by a native speaker. |
||
|
5. Additional clarifications |
||

Reviewer 2 Report
Comments and Suggestions for Authors
The premise of this study was to assess multi-dimensional chronotypes of morning and evening type states and traits among young adult college students and associations with health, mood, and sleep. Simulations were conducted to evaluate for differences between concordant and discordant types with comparisons between weekday and weekend self-reports of sleep activity.
This is an interesting manuscript that adds knowledge to associations between sleep and health.
The authors appropriately noted the limitations of the study, particularly in this being a cross-study design. Other limitations not identified include characteristics that may influence sleep in this student population (e.g., preparing for examinations/other schoolwork; pre-existing health conditions; home environmental factors; health behaviors [e.g., smoking, alcohol consumption, engagement in physical activity, diet]). In addition, analyzing various clock genes and/or calculating epigenetic age acceleration would be a more accurate method of assessing biological clock differences in relation to sleep states/traits and associations with health. Furthermore, this heterogenous sample is not generalizable to populations across age and education levels. It would be helpful to have these limitations addressed in the discussion section.
Overall, this was an interesting and potentially informative study. A future study that evaluated sleep patterns longitudinally (preferably with the aid of wearable sleep tracker), including multiple characteristics that may influence sleep patterns, and measuring genetic/epigenetic markers would enhance this area of research substantially.
Author Response
|
Response to Reviewer #2 Comments
|
||
|
1. Summary |
|
|
|
Please find the detailed responses below and the corresponding revisions/corrections highlighted in the re-submitted files in red.
|
||
|
2. Questions for General Evaluation |
Reviewer’s Evaluation |
Response and Revisions |
|
Does the introduction provide sufficient background and include all relevant references? |
Yes/ |
|
|
Are all the cited references relevant to the research? |
Yes/ |
|
|
Is the research design appropriate? |
Yes/ |
|
|
Are the methods adequately described? |
/Not applicable |
|
|
Are the results clearly presented? |
Yes/ |
|
|
Are the conclusions supported by the results? |
Yes/ |
|
|
3. Point-by-point response to Comments and Suggestions for Authors |
||
|
Comments 1: The authors appropriately noted the limitations of the study, particularly in this being a cross-study design. Other limitations not identified include characteristics that may influence sleep in this student population (e.g., preparing for examinations/other schoolwork; pre-existing health conditions; home environmental factors; health behaviors [e.g., smoking, alcohol consumption, engagement in physical activity, diet]). In addition, analyzing various clock genes and/or calculating epigenetic age acceleration would be a more accurate method of assessing biological clock differences in relation to sleep states/traits and associations with health. Furthermore, this heterogenous sample is not generalizable to populations across age and education levels. It would be helpful to have these limitations addressed in the discussion section. |
||
|
Response 1: Thank you for pointing this out. We agree with this comment. Therefore, we included these limitations in the last paragraph of Discussion. |
||
|
Comments 2: Overall, this was an interesting and potentially informative study. A future study that evaluated sleep patterns longitudinally (preferably with the aid of wearable sleep tracker), including multiple characteristics that may influence sleep patterns, and measuring genetic/epigenetic markers would enhance this area of research substantially. |
||
|
Response 1: Thank you for this evaluation. These recommendations for future studies were also included the limitation paragraph of Discussion. 4. Response to Comments on the Quality of English Language |
||
|
Point 1: The English is fine and does not require any improvement. |
||
|
Response 1: Thank you very much indeed for this evaluation, and for the previous comments and recommendations that helped to revise and improve the manuscript. |
||
|
5. Additional clarifications |
||

Reviewer 3 Report
Comments and Suggestions for Authors
The authors describe a study where state and trait assessments of chronotype are contrasted to see if there is more to sleep timing than can be explained by owls and larks. This is interesting and important work, but the quality of the presentation of the Introduction, Results, and Methods must be improved.
Introduction
-The numbered questions imbedded in paragraphs one and two are very difficult to read and interrupt the flow of the writing. Please consider changing the formatting or rewriting these paragraphs to explain the background in greater detail before introducing the research questions.
-"Since the correlations between trait- and state-assessments are usually positive in sign and substantial in magnitude, it is difficult to determine whether the established associations either internal or external in nature and, therefore, it is difficult to answer to the question about the prospects of correction of such state-like characteristic of an individual as sleep timing for modification of the pattern of association of chronotype with health, mood, and sleep. Light on such issue can be shed in a study of individuals with contradictive trait and state assessments of morningness-eveningness in natural settings." While this paragraph is poetic, it is difficult to understand what the authors are trying to say. I would like more information about previously discovered correlations between trait and state assessments, with more introduction to the nature of the relationship than just whether it was a positive or negative and weak or strong correlation. I want to know more about how previous findings indicate a knowledge gap with regards to chronotype, health, mood, and sleep. Define what constitutes "a contradictive trait and state assessment".
-"University students are similar to people of younger and older ages in the obligation to wake-up early on weekdays". Please actually describe this obligation rather than referring to it in vague terms.
-The difference between morningness-eveningness chronotypes and LIVEMAN types should be explained in the Introduction. LIVEMAN determination should also be described in the Methods section.
-The closing paragraph of the Introduction mixes imbedded numbered questions with indented numbered hypotheses. Please choose a consistent format that makes it easy for the reader to understand the study questions and hypotheses.
Results
-Demographics?
-There is no clear explanation about how students were assigned to either discordant groups or the concordant group. This information should be included in the Methods in detail. Since the Results are presented before the Methods, the authors should make sure the logic behind the groupings is clear right away.
-A diagram or table explaining the conceptual difference between the kinds of discordant types observed in this study would go a long way in helping the reader understand the issue. It is difficult to imagine college students who wake up earlier on the weekends. Including the number of students who fell into each discordant group as part of a conceptual overview would be particularly helpful.
Figure 1B makes it appear that all groups are waking earlier on the weekdays than on the weekends. What is the difference between groups then?
-Tables 1 and 2 need to be organized better. Each table appears to be two tables mushed together. The title rows are confusing.
-There is an extra t in Table 1: "Ttrait+state type"
-Where is the ratio between expected and real count of university students written in Table 1?
-The modeling parameters are not clearly explained in the context of the study. At first glance, it appears that the extra parameters are just additional math without relevance to health outcomes. "Indeed, the model without the additional assumption accurately simulated sleep times in three trait+state types... However, two other trait+state types (Me- and Em-types 306
with discordant trait- and state-characteristics) were the exceptions". You cannot see this graphically.
Without more information about how the students were divided into subgroups, I cannot offer any more detailed advice about the Results section.
Discussion
-In what way was "biologically determined bedtimes" measured in this survey study? Is this an assumption based on the model?
-"...results of present model-based simulations suggested that bedtimes in, at least, one of 5 trait+state subsamples cannot be explained by the model without the additional assumption that sleep can be partially self-deprived in this subsample due to the conflict between the biological and “mental” clocks." There has to be a way to show this clearly in the figures.
-The Discussion is well written and explains what is frustratingly opaque in the Results and Methods sections. There are, unfortunately, many typos. Also, people do not refer to themselves as "your organism". I'm not sure what the authors are trying to say.
Methods
-Where is the information about the LIVEMAN chronotype test?
-The Methods section on "Statistical Analysis" title is misspelled.
-There needs to be more information about what statistical tests were performed for this study and how groups were determined.
Conclusions
-The conclusion section is very well-written. I would like this kind of clarity in writing extended to the entire manuscript.
Comments on the Quality of English Language
Please proofread the manuscript for typos throughout.
While I find the authors' writing style to be beautiful, it is not conducive to understanding the purpose of the study. I recommend substituting the flowery language for concise explanations when writing scientific articles. Please continue to write beautifully for other essays, poetry, or fiction.
Author Response
Response to Reviewer #3 Comments
|
||
|
1. Summary |
|
|
|
Please find the detailed responses below and the corresponding revisions/corrections highlighted in the re-submitted files in red.
|
||
|
2. Questions for General Evaluation |
Reviewer’s Evaluation |
Response and Revisions |
|
Does the introduction provide sufficient background and include all relevant references? |
Can be improved/Must be improved/Not applicable |
The introduction section was rewritten |
|
Are all the cited references relevant to the research? |
/Must be improved/Not applicable |
The list of references was doubled |
|
Is the research design appropriate? |
/Must be improved/Not applicable |
The result section was revised and, in particular, restructured to introduce more details on the methodology before the method section |
|
Are the methods adequately described? |
/Can be improved/Must be improved/Not applicable |
The method section was extended and revised |
|
Are the results clearly presented? |
Yes/Can be improved/Must be improved/Not applicable |
|
|
Are the conclusions supported by the results? |
/Can be improved/Must be improved/Not applicable |
The conclusion section was revised |
|
3. Point-by-point response to Comments and Suggestions for Authors |
||
|
Comments 1: The authors describe a study where state and trait assessments of chronotype are contrasted to see if there is more to sleep timing than can be explained by owls and larks. This is interesting and important work, but the quality of the presentation of the Introduction, Results, and Methods must be improved. |
||
|
Response 1: Thank you for pointing this out. We agree with this recommendation, and we tried to improve the entire manuscript by revision of the introduction, results, and methods sections. |
||
|
Comments 2: Introduction -The numbered questions imbedded in paragraphs one and two are very difficult to read and interrupt the flow of the writing. Please consider changing the formatting or rewriting these paragraphs to explain the background in greater detail before introducing the research questions. |
||
|
Response 2: The numbered questions of these paragraphs were revised, the numeration was excluded, the numeration remained exclusively for the study hypotheses of the last paragraph of Introduction. Comments 3: Introduction -"Since the correlations between trait- and state-assessments are usually positive in sign and substantial in magnitude, it is difficult to determine whether the established associations either internal or external in nature and, therefore, it is difficult to answer to the question about the prospects of correction of such state-like characteristic of an individual as sleep timing for modification of the pattern of association of chronotype with health, mood, and sleep. Light on such issue can be shed in a study of individuals with contradictive trait and state assessments of morningness-eveningness in natural settings." While this paragraph is poetic, it is difficult to understand what the authors are trying to say. I would like more information about previously discovered correlations between trait and state assessments, with more introduction to the nature of the relationship than just whether it was a positive or negative and weak or strong correlation. I want to know more about how previous findings indicate a knowledge gap with regards to chronotype, health, mood, and sleep. Define what constitutes "a contradictive trait and state assessment". Response 3: This paragraph was fully rewritten, new paragraphs were added in Introduction, including more details on earlier established correlations, including their strength and direction. More references were added on the associations of chronotype with health, mood, and sleep. And "a contradictive trait and state assessment" was explained. Comments 4: Introduction -"University students are similar to people of younger and older ages in the obligation to wake-up early on weekdays". Please actually describe this obligation rather than referring to it in vague terms. Response 4: This paragraph was fully rewritten. Comments 5:. Introduction -The difference between morningness-eveningness chronotypes and LIVEMAN types should be explained in the Introduction. LIVEMAN determination should also be described in the Methods section. Response 5: The difference between morningness-eveningness chronotypes and LIVEMAN types was explained in Abstract and other sections, including a more detailed description of these types in Results and Methods. Comments 6:. Introduction -The closing paragraph of the Introduction mixes imbedded numbered questions with indented numbered hypotheses. Please choose a consistent format that makes it easy for the reader to understand the study questions and hypotheses. Response 6: The introduction section was enlarged and rewritten, including the objectives and hypotheses of the study. Comments 7:. Results -Demographics? Response 7: The results of calculation of ages of male and female students were included in the first paragraph of Results, and, in the following paragraphs, the results of the effect of sex on chronotyping were included. Comments 8:. Results There is no clear explanation about how students were assigned to either discordant groups or the concordant group. This information should be included in the Methods in detail. Since the Results are presented before the Methods, the authors should make sure the logic behind the groupings is clear right away. Response 8: Thank you for pointing this out. We agree with this recommendation, and additional information was included in Results and Methods, in particular, new table (Table 1 of the revised version) and the notes to this table were added to provide the detailed description of students’ assignment into subsamples with different trait and state assessments. Comments 9:. Results -A diagram or table explaining the conceptual difference between the kinds of discordant types observed in this study would go a long way in helping the reader understand the issue. It is difficult to imagine college students who wake up earlier on the weekends. Including the number of students who fell into each discordant group as part of a conceptual overview would be particularly helpful. Response 9: See the previous reply. We did exactly what was recommended, thank you, and we enjoyed the phrase “It is difficult to imagine college students who wake up earlier on the weekends” so much that stolen it for improving Introduction. Comments 10:. Results Figure 1B makes it appear that all groups are waking earlier on the weekdays than on the weekends. What is the difference between groups then?. Response 10: It is not surprising that all groups waked up earlier on weekdays, because they attended the same classes. Despite that, a relatively small difference in weekday risetimes between the extreme types was found to be significant (Table 3). Comments 11:. Results -Tables 1 and 2 need to be organized better. Each table appears to be two tables mushed together. The title rows are confusing.. Response 11: We agree with this recommendation, and we divided these two tables. Now these are four tables (Tables 2-5 of the revised version). The titles were also corrected. Comments 12:. Results -There is an extra t in Table 1: "Ttrait+state type" Response 12: Thank you for pointing this out. This and several other typos were corrected. Comments 13:. Results -Where is the ratio between expected and real count of university students written in Table 1? Response 13: We rewrite the title and notes to this table to explain the way of calculating this ratio. Comments 14:. Results -The modeling parameters are not clearly explained in the context of the study. At first glance, it appears that the extra parameters are just additional math without relevance to health outcomes. "Indeed, the model without the additional assumption accurately simulated sleep times in three trait+state types... However, two other trait+state types (Me- and Em-types with discordant trait- and state-characteristics) were the exceptions". You cannot see this graphically. Response 14: We highlighted this extra parameter in the table, explained it in the notes to this table and in the legend to Figure 2B that shows for Me- and Em-types (and it is absent in other types, e.g., Figure 2A). Comments 15:. Results Without more information about how the students were divided into subgroups, I cannot offer any more detailed advice about the Results section. " Response 15: See Response 8: We added additional information in Results and Methods, in particular, we added new table (Table 1 of the revised version) that explains the division into subsamples. Comments 16:. Discussion -In what way was "biologically determined bedtimes" measured in this survey study? Is this an assumption based on the model? Response 16: Yes, it is impossible to find out - from data on bedtimes - when "biologically determined bedtimes" occurred. We can only conclude that – at least in some od subsamples – they occurred before the reported bedtimes. Therefore, the simulations were necessary to include in the paper to demonstrate that bedtimes in some of the subsamples cannot be explained without this assumption (i.e., when - in one of the subsamples - weekday bedtimes were reported to occur later than weekend bedtimes). Comments 17:. Discussion -"...results of present model-based simulations suggested that bedtimes in, at least, one of 5 trait+state subsamples cannot be explained by the model without the additional assumption that sleep can be partially self-deprived in this subsample due to the conflict between the biological and “mental” clocks." There has to be a way to show this clearly in the figures. Response 17: See Response 14. Comments 18:. Discussion --The Discussion is well written and explains what is frustratingly opaque in the Results and Methods sections. There are, unfortunately, many typos. Also, people do not refer to themselves as "your organism". I'm not sure what the authors are trying to say. Response 18: Thank you for this evaluation of the discussion section. Thank you for pointing on the typos, the typos were corrected in this and other sections. The phrase containing "your organism" was excluded. Instead, the paragraph of with limitations and future research aims was extended. Comments 19:. Methods -Where is the information about the LIVEMAN chronotype test? Response 19: More information was provided for the SIC both in the methods and results sections. Comments 20:. Methods -The Methods section on "Statistical Analysis" title is misspelled Response 20: Honestly, we do not understand what “is misspelled” in "Statistical Analysis". We even searched in Google tried to find out what this phrase can mean: “The Methods section on "Statistical Analysis" title is misspelled” (https://www.google.com/search?q=The+Methods+section+on+%22Statistical+Analysis%22+title+is+misspelled&oq=The+Methods+section+on+%22Statistical+Analysis%22+title+is+misspelled&gs_lcrp=EgZjaHJvbWUyBggAEEUYOdIBCTMwMDFqMGoxNagCCLACAQ&sourceid=chrome&ie=UTF-8). Here is the result of this search “AI Overview: The title "Statistical Analysis" in a Methods section may be a misspelling or a less descriptive title. It's more common and accurate to use titles like "Data Analysis," "Statistical Methods," or "Analytical Techniques" within the Methods section when describing how statistical methods were employed in a study.”. Therefore, we changed "Statistical Analysis" on “Statistical Methods” in the title of the subsection of Methods. Comments 21:. Methods -There needs to be more information about what statistical tests were performed for this study and how groups were determined Response 21: All statistical tests are briefly described in this subsection, and in the notes to tables with the results of these tests. Comments 22:. Conclusions -The conclusion section is very well-written. I would like this kind of clarity in writing extended to the entire manuscript Response 22: Thank you very much indeed for this evaluation and for all previous detailed comments and recommendations that helped to revise and improve the manuscript. We tried to do our best. |
||
|
4. Response to Comments on the Quality of English Language |
||
|
Point 1: Please proofread the manuscript for typos throughout. While I find the authors' writing style to be beautiful, it is not conducive to understanding the purpose of the study. I recommend substituting the flowery language for concise explanations when writing scientific articles. Please continue to write beautifully for other essays, poetry, or fiction |
||
|
Response 1: Unfortunately or fortunately, there is no time to write anything but scientific papers, might be with the obituaries as the only exception, e.g.: https://www.tandfonline.com/doi/full/10.1080/09291016.2024.2418941?src=most-read-last-year |
||
|
5. Additional clarifications |
||

Round 2
Reviewer 1 Report
Comments and Suggestions for Authors
The authors should explain in detail how the tools they used relate to the MEQ and the MCTQ, which are aimed at improving here, but apparently not used in the present study. Also, related to this, both the MEQ and the MCTQ assess each continuous outcomes, and not dichotomous outcomes, for statistical purposes often groups a built, but neither tool provides grouped outcomes. Hence the argument these tools only assess early and late groups is too strict and not true. The manuscript as submitted is difficult to review without knowing the exact questions asked, meaning the tools used should be submitted as well. It is difficult to understand what a mental clock should be, it's the noise in the data and the behavioral parts in the end, what you see is the interindividual variance in how people can cope with sleep pressure, hence a measure of the homeostat and not a "mental clock". Last, not least, the title should be changed together with the use of the terms "larks" and "owls", terms which should be avoided in this context, its early types etc. but not an animal.
Comments on the Quality of English LanguageThat manuscript is difficult to read, lacks a red line, needs some re-writing.
Author Response
|
Response to Reviewer #1 Comments (the second round)
|
||
|
1. Summary |
|
|
|
Please find the detailed responses below and the corresponding revisions/corrections highlighted in the re-submitted files in red.
|
||
|
2. Questions for General Evaluation |
Reviewer’s Evaluation |
Response and Revisions |
|
Does the introduction provide sufficient background and include all relevant references? |
Must be improved/Not applicable |
The introduction section was improved, including the list the study questions. |
|
Are all the cited references relevant to the research? |
Must be improved/Not applicable |
The list of references was doubled in the previous revision, and 10 more references were added. |
|
Is the research design appropriate? |
Must be improved/Not applicable |
All manuscript sections were further revised and improved. |
|
Are the methods adequately described? |
Can be improved/Must be improved/Not applicable |
The method section was further extended. The main questionnaire was included in Supplementary Materials, the English versions of other questionnaires can be found in the cited original journal publications. |
|
Are the results clearly presented? |
Can be improved/Must be improved/Not applicable |
The result section was further improved. |
|
Are the conclusions supported by the results? |
Can be improved/Must be improved/Not applicable |
The conclusion section was further revised. |
|
3. Point-by-point response to Comments and Suggestions for Authors |
||
|
Comments 1: The authors should explain in detail how the tools they used relate to the MEQ and the MCTQ, which are aimed at improving here, but apparently not used in the present study. |
||
|
Response 1: The method section was extended. The differences between morningness-eveningness chronotypes and LIVEMAN types was explained in several sections of the manuscript, including a more detailed description of these types in Results and Methods. All questionnaire tools were previously validated against traditional tools for assessment of morningness-eveningness and sleep times on weekdays and weekends. |
||
|
Comments 2: Also, related to this, both the MEQ and the MCTQ assess each continuous outcomes, and not dichotomous outcomes, for statistical purposes often groups a built, but neither tool provides grouped outcomes. Hence the argument these tools only assess early and late groups is too strict and not true. |
||
|
Response 2: The new table (Table 1) and the notes to this table were added to the previously revised version to provide the detailed description of method of students’ assignment to subsamples with different trait and state assessments of morningness-eveningness. In particular, the dichotomous outcomes of the questionnaires are explained in the notes to this Table 1. More details were provided for the morningness-eveningness scales of two questionnaires in Methods (e.g., that these questionnaires were previously validated not only against several scales of other morningness-eveningness questionnaires, but also against objective markers of the circadian phase, daily rhythms of objective (e.g., spectral EEG) indexes of sleepiness, etc. Some of such references, 10 in total, were added. Comments 3: The manuscript as submitted is difficult to review without knowing the exact questions asked, meaning the tools used should be submitted as well. Response 3: Four questions were highlighted and improved in Introduction to assign them to four hypotheses. The SIC is included in Supplementary Materials in English and Russian, the lists of items of other chronobiological questionnaires were previously included in their cited journal publications. Comments 4: It is difficult to understand what a mental clock should be, it's the noise in the data and the behavioral parts in the end, what you see is the interindividual variance in how people can cope with sleep pressure, hence a measure of the homeostat and not a "mental clock". Response 4: All paragraphs about mental clocks were either excluded (e.g., 2.6.“Mental Clocks” of Results) or rewritten (Discussion and Conclusion). In the revised version, we refer to earlier invented term “mind-bedtime procrastination” instead of “mental clocks”. Comments 5: Last, not least, the title should be changed together with the use of the terms "larks" and "owls", terms which should be avoided in this context, its early types etc. but not an animal. Response 5: In subtitle, larks and owls were replaced by morning and evening types. Almost all mentioning owls and larks were also excluded from entire test. The title is the citation, and, in that context, it is not about animals (see 5. Additional clarifications).
|
||
|
4. Response to Comments on the Quality of English Language |
||
|
Point 1: That manuscript is difficult to read, lacks a red line, needs some re-writing. |
||
|
Response 1: English was improved everywhere in the manuscript, and the previous revised version was checked by a native speaker. |
||
|
5. Additional clarifications About the title: see pdf file |
||

Reviewer 3 Report
Comments and Suggestions for Authors
This manuscript reports an ambitious attempt to quantify chronotype in a student population with more nuance than simple morningness/eveningness and next, to model-based simulations of sleep-wake cycles in discordant subsamples. My overall recommendation is for the authors to divide the content presented here into two (or more) manuscripts so that they may dedicate enough analysis and explanation to each study question. Study questions 3 and 4 are contingent upon results of questions 1 & 2, so could easily be moved to a second paper. Instead of cramming in results, the authors should focus on more clearly explaining their methodology and the results that they received at each step.
Introduction
-The authors need to explain the differences between the LIVEMAN and MEQ questionnaires in the introduction. The use of this state/trait questionnaire is pivotal to the study design but is not introduced. The author should dedicate more time in the paper to explaining the LIVEMAN chronotypes and the background of this item.
-Research questions: 1. "How many these chronotypes among university students? and 2. Can they belong to other than morning or evening types? are the same question phrased slightly differently."
-Research question 3. "Can they resemble morning and evening types on the patterns of association with health, mood, and sleep concerns?" is unclear and contingent on the expectation that students will not adhere to morning and evening types (question 2).
4. Can model-based simulations of their sleep-wake cycles link the conflicting trait- and state-characteristics with the parameters of underlying processes of sleep-wake
regulation?
Results
-The Table notes are extensive, and yet, the reader cannot understand the table without this information. Move these explanations to the Introduction, Methods or Results in-text section.
-The Figure 1 caption does not explain what the graph shows.
Discussion
-The idea of adding a social clock regulator to a biomathematical model of sleep-wake cycle is very intriguing. Rather than squeezing the model simulation results into this paper that also focuses on survey results, I would like to see the authors explain the logic for the social regulator in greater depth in its own paper that builds upon the survey findings.
-The conclusion proposes an interesting theory about social clocks but does not really tie back in to study questions 1 through 3. What did the survey findings tell us about how multidimensional chronotyping depicts student sleep-wake cycles relative to MEQ? What about the health, mood, and sleep concerns? The Conclusion is detached from the Discussion section.
Author Response
Response to Reviewer #3 Comments
|
||
|
1. Summary |
|
|
|
Please find the detailed responses below and the corresponding revisions/corrections highlighted in the re-submitted files in red.
|
||
|
2. Questions for General Evaluation |
Reviewer’s Evaluation |
Response and Revisions |
|
Does the introduction provide sufficient background and include all relevant references? |
/Must be improved/Not applicable |
The introduction section was extended to meet some of the recommendations |
|
Are all the cited references relevant to the research? |
/Can be improved/Must be improved/Not applicable |
Several new – most recent - references were added |
|
Is the research design appropriate? |
/Must be improved/Not applicable |
The research design was additionally clarified by adding the last subsection of Results, by explaining Table 1 in the first paragraphs of Results, by adding several new sentences to other subsections of Results, and to Discussion and Conclusions |
|
Are the methods adequately described? |
/Must be improved/Not applicable |
The methodology of chronotyping was presented more clearly in Results concerning Table 1 and some other paragraphs of Results |
|
Are the results clearly presented? |
Yes/Can be improved/Must be improved/Not applicable |
|
|
Are the conclusions supported by the results? |
/Can be improved/Must be improved/Not applicable |
The conclusion section was extended to meet the recommendations |
|
3. Point-by-point response to Comments and Suggestions for Authors |
||
|
Comments 1: This manuscript reports an ambitious attempt to quantify chronotype in a student population with more nuance than simple morningness/eveningness and next, to model-based simulations of sleep-wake cycles in discordant subsamples. My overall recommendation is for the authors to divide the content presented here into two (or more) manuscripts so that they may dedicate enough analysis and explanation to each study question. Study questions 3 and 4 are contingent upon results of questions 1 & 2, so could easily be moved to a second paper. Instead of cramming in results, the authors should focus on more clearly explaining their methodology and the results that they received at each step. |
||
|
Response 1: We tried to further improve all sections of the manuscript (Introduction, Results, Discussion, Methods, and Conclusions) to stress the close interrelationship between 4 study questions. In particular, several questions were slightly corrected and special subsection of the result section (the last one) was added to show that the answer to the first question rises the next 2nd question, the answer to the second question rises the 3rd question, etc. until to the answer to the 4th question explaining the results of empirical study in terms of the model of sleep-wake regulation (see also the below replies). |
||
|
Comments 2: Introduction -The authors need to explain the differences between the LIVEMAN and MEQ questionnaires in the introduction. The use of this state/trait questionnaire is pivotal to the study design but is not introduced. The author should dedicate more time in the paper to explaining the LIVEMAN chronotypes and the background of this item. |
||
|
Response 2: As recommended, the difference between LIVEMAN and MEQ questionnaires was briefly explained in Introduction. The implementation of trait and state assessments in the present study was explained in several new paragraphs in the beginning of Results in more details. In one of the following subsections of the result section, a short description of LIVEMAN types was added. Results on LIVEMAN types were also addressed in several new sentences of Discussion. Comments 3: Introduction -Research questions: 1. "How many these chronotypes among university students? and 2. Can they belong to other than morning or evening types? are the same question phrased slightly differently." Response 3: To show that these are the two different questions with two different answers to them, we made new (the last) subsection of Results with new table and included this table these answers. Comments 4: Introduction -Research question 3. "Can they resemble morning and evening types on the patterns of association with health, mood, and sleep concerns?" is unclear and contingent on the expectation that students will not adhere to morning and evening types (question 2). Response 4: Similarly, we included this table these answers to these questions. We also added a sentence in Introduction to exemplify the expectation that students will not adhere to morning and evening types and, therefore, they do not show the association with health, mood, and sleep problems reported for these types. Finally, we illustrated the relationship between these two questions in Discussion by referring to one of the most current publications on the association of the chronotypes suggested by the SIC with mental health. Comments 5:. Introduction 4. Can model-based simulations of their sleep-wake cycles link the conflicting trait- and state-characteristics with the parameters of underlying processes of sleep-wake regulation?. Response 5: We corrected to make clearer this question and its hypothesis, and showed the relationship of this question to the three questions of the questionnaire study. We added new sentence in the beginning of simulation subsection of Results to show the difference between the present and previous model-based simulations, we added the final paragraph in this subsection to explain the relationship between the simulation and empirical results of the present study, and we added a new sentence in Conclusions to stress the difference between the present and previous model-based simulations that provides the explanation of conflicting trait- and state-characteristics in terms of the model. Comments 6:. Results -The Table notes are extensive, and yet, the reader cannot understand the table without this information. Move these explanations to the Introduction, Methods or Results in-text section. Response 6: As recommended, this Table was explained in the beginning of Results. Comments 7:. Results -The Figure 1 caption does not explain what the graph shows Response 7: The caption was extended to explain what the graphs show. Comments 8:. Discussion -The idea of adding a social clock regulator to a biomathematical model of sleep-wake cycle is very intriguing. Rather than squeezing the model simulation results into this paper that also focuses on survey results, I would like to see the authors explain the logic for the social regulator in greater depth in its own paper that builds upon the survey findings. Response 8: As we mentioned in Reply 5, we corrected to make clearer this simulation question and its hypothesis, and we showed the relationship of this question to the three questions of the questionnaire study. The concept of social regulator was discussed in a new paragraph of Discussion concerning the possible aims of future research. Comments 9: -The conclusion proposes an interesting theory about social clocks but does not really tie back in to study questions 1 through 3. What did the survey findings tell us about how multidimensional chronotyping depicts student sleep-wake cycles relative to MEQ? What about the health, mood, and sleep concerns? The Conclusion is detached from the Discussion section.. Response 9: As we mentioned in Reply 4, we illustrated the relationship between the first questions in Discussion by referring to one of the most current publications on the association of the chronotypes suggested by the SIC with mental health. The relationships between the simulation results and the results on health, mood, and sleep concerns were described in a new paragraph of the subsection of simulation results. See also the previous replies.
4. Response to Comments on the Quality of English Language |
||
|
Point: The English is fine and does not require any improvement |
||
|
Response: Thank U. |
||
|
5. Additional clarifications None |
||

Round 3
Reviewer 1 Report
Comments and Suggestions for Authors
The Supplementary Materials are not in good shape, the English and Russian parts should be presented in a separate way.
Figures 1 and 2 need to be tidied, especially Figure 2 with its distorted time line (x-axis).
Argumentation in the manuscript text:
At several points the authors claim or argue, that the conventional tools to assess chronotype would only allow to either classify a person as an early type or a late type. E.g. on page 2 it says "However, the discrepancy between trait and state assessments of morningness- 92
eveningness can be a consequence of applying the unidimensional approach to categori- 93
zation of chronotypes, i.e., when only two - either evening or morning – types are distin- 94
guished. The application of multidimensional approach to chronotyping can show that 95
individuals with discordant trait and state assessments belong to other than just two - 96
morning and evening - types. Consequently, these other chronotypes can differ from 97
morning and evening types in the association of their chronotype with health, mood, and 98
sleep problems." However, this is not the case, since the MCTQ and MEQ both result on a continuous scale. And, yes, the variance along this continuous measures is of interest, and one can call it a multidimensional approach, but the argument is wrong in the sense to claim that a priori only two poles would exist. This should be clarified in the manuscript text, meaning to strengthen the rationale of the paper.
Related to this, the concept of circadian typology (different from the concept of chronotype) should be discussed in more depth, also he concept of social jetlag appears a little neglected in the discussion.
Comments on the Quality of English LanguageThe manuscript text still requires language editing.
Author Response
|
Response to Reviewer #1 Comments (the second round)
|
||
|
1. Summary |
|
|
|
Please find the detailed responses below and the corresponding revisions/corrections highlighted in the re-submitted manuscript in red.
|
||
|
2. Questions for General Evaluation |
Reviewer’s Evaluation |
Response and Revisions |
|
Does the introduction provide sufficient background and include all relevant references? |
Can be improved/Must be improved/Not applicable |
The introduction section was improved and further extended. |
|
Are all the cited references relevant to the research? |
Can be improved/Must be improved/Not applicable |
The list of references was extended. |
|
Is the research design appropriate? |
Can be improved/Must be improved/Not applicable |
All manuscript sections were further revised and improved. |
|
Are the methods adequately described? |
/Must be improved/Not applicable |
The method of chronotyping and its results were described in details in the result section. A short description of the SIC was included in Introduction as recommended. Supplementary Material with its illustration was revised to show separately the English and Russian versions. |
|
Are the results clearly presented? |
Yes/Can be improved/Must be improved/Not applicable |
|
|
Are the conclusions supported by the results? |
Can be improved/Must be improved/Not applicable |
The conclusion section was extended as recommended. |
|
3. Point-by-point response to Comments and Suggestions for Authors |
||
|
Comments 1: The Supplementary Materials are not in good shape, the English and Russian parts should be presented in a separate way. |
||
|
Response 1: Supplementary Material with the illustration of the SIC was revised to show separately its English and Russian versions. |
||
|
Comments 2: Figures 1 and 2 need to be tidied, especially Figure 2 with its distorted time line (x-axis). |
||
|
Response 2: The x-axis of each of the figures (Figures 1A, 1B, 2A and 2B) were modified to make them clearer and to improve their visibility for a reader. Comments 3: Argumentation in the manuscript text: At several points the authors claim or argue, that the conventional tools to assess chronotype would only allow to either classify a person as an early type or a late type. E.g. on page 2 it says "However, the discrepancy between trait and state assessments of morningness- 92eveningness can be a consequence of applying the unidimensional approach to categori- 93zation of chronotypes, i.e., when only two - either evening or morning – types are distin- 94guished. The application of multidimensional approach to chronotyping can show that 95individuals with discordant trait and state assessments belong to other than just two - 96morning and evening - types. Consequently, these other chronotypes can differ from 97morning and evening types in the association of their chronotype with health, mood, and 98sleep problems." However, this is not the case, since the MCTQ and MEQ both result on a continuous scale. And, yes, the variance along this continuous measures is of interest, and one can call it a multidimensional approach, but the argument is wrong in the sense to claim that a priori only two poles would exist. This should be clarified in the manuscript text, meaning to strengthen the rationale of the paper. Response 3: A couple of sentences were added in the first paragraph of Introduction to clarify how the continuous scales are usually used for distinguishing between two extreme chronotypes. If this is a single dimension, it has only two poles, and, therefore, it allows the distinguishing between or categorization into only two such distinct chronotypes. This issue was also clarified in other parts of the manuscript. Comments 4: Related to this, the concept of circadian typology (different from the concept of chronotype) should be discussed in more depth, also he concept of social jetlag appears a little neglected in the discussion. Response 4: First half of the comment. We added this discussion to one of the paragraphs of Discussion. However, it has to be emphasized here that the terms “circadian type” and “chronotype” are mostly used interchangeably in the literature (eg, see the title of ref [13] with the first author who was the first to introduce the term “chronotype” in its recent meaning – see ref [16]). Therefore, in this additional discussion, we pointed at those publications in which a wider interpretation of these two terms was used (eg, ref [4,5,7,8,14,15]). Second half of the comment. The authors of the term and concept of social jetlag (ref [72]) introduced the idea of the conflicting clocks, biological and social, that, as we showed in this manuscript, can be further elaborated by the including an additional conflict with what can be called “mind-bedtime procrastination” or “mental clocks”. As for the mechanism suggested by these authors for social jetlag (the circadian phase does not change during transitions between weekdays and free days being controlled by the sun clocks, while the sleep phase is shifted relative to this unchanged circadian phase), we failed to support it in the previous simulations of sleep times (Putilov AA. Reaction of the endogenous regulatory mechanisms to early weekday wakeups: a review of its popular explanations in light of model-based simulations. Front Netw Physiol. 2023 Dec 15;3:1285658. doi: 10.3389/fnetp.2023.1285658) and we showed in the latest publication (Putilov AA et al. Estimation of sleep shortening and sleep phase advancing in response to advancing risetimes on weekdays. Chronobiol Int. 2025 Jun;42(6):770-783. doi: 10.1080/07420528.2025.2509629) and we will show in the recent submission to another journal that the circadian phase is shifting ahead in response to shifting the weekday wake-ups, and that this shift cannot be interpreted as the shift of the sleep phase relative to the phase of the circadian clocks.
|
||
|
4. Response to Comments on the Quality of English Language |
||
|
Point: The manuscript text still requires language editing. |
||
|
Response 1: English was further improved everywhere in the manuscript, and the previous revised version was checked by a native speaker. |
||
|
5. Additional clarifications None |
||

Reviewer 3 Report
Comments and Suggestions for Authors
The authors have satisfied the reviewers' requests. The manuscript has been greatly improved and is ready for publication.
Author Response
Reply to Reviewer #2.
Open Review
(x) I would not like to sign my review report
( ) I would like to sign my review report
Quality of English Language
( ) The English could be improved to more clearly express the research.
(x) The English is fine and does not require any improvement.
Comments and Suggestions for Authors
The authors have satisfied the reviewers' requests. The manuscript has been greatly improved and is ready for publication.
Reply. Thank you.

Round 4
Reviewer 1 Report
Comments and Suggestions for Authors
Thank you for addressing all points raised during the review process.
Comments on the Quality of English LanguageI suggest the journal team to help improve the writing.
Author Response
Reply to Reviewer #1.
Open Review
(x) I would not like to sign my review report
( ) I would like to sign my review report
Quality of English Language
(x) The English could be improved to more clearly express the research.
( ) The English is fine and does not require any improvement.
Comments and Suggestions for Authors
Thank you for addressing all points raised during the review process.
Reply. Thank you2.
Comments on the Quality of English Language
I suggest the journal team to help improve the writing.
Reply. The English was improved to more clearly express the research, and Ana K. Jones, a native speaker from the Institute of English and American Studies of the Humboldt University (Berlin), edited the revised version.
( ) The English could be improved to more clearly express the research.
(x) The English is fine and does not require any improvement.
